# Prevalence of nasopharyngeal Streptococcus *Pneumoniae* carriage in infants: A systematic review and meta-analysis of cohort studies and randomized controlled trials

Gulzhan Beissegulova[1]*, Bakyt Ramazanova[1], Kamilya Mustafina[1], Tolkyn Begadilova[1], Yekaterina Koloskova[1], Bibigul Seitkhanova[2], Aliya Mamatova[1], Ulzhan Iskakova[1], Ratbek Sailaubekuly[2], Zhaksylyk Seiitbay[3]

1 Department of Microbiology and Virology, Asfendiyarov Kazakh National Medical University, Almaty, Kazakhstan, 2 Department of Microbiology, Virology and Immunology, South Kazakhstan Medical Academy, Shymkent, Kazakhstan, 3 School of General Medicine-2, Asfendiyarov Kazakh National Medical University, Almaty, Kazakhstan

* g.beisegulova@kaznmu.kz

## Abstract

This study aims to examine the prevalence of nasopharyngeal Streptococcus pneumoniae carriage (NSPC) in infants during their first two years of life and to compare the carriage rates among different vaccine groups and country income-levels. This will be achieved through a systematic review of the published literature, specifically focusing on data from cohort studies and randomized controlled trials. A comprehensive search was conducted in four electronic databases: PubMed, Web of Science, ScienceDirect, and Scopus, using a predefined search strategy. Forty-nine articles met the inclusion criteria for this systematic review. According to the results obtained from the random effects model, the pooled mean prevalence of NSPC was 1.68% at birth (95% CI [0.50; 5.47]), 24.38% at 1 to 4 months (95% CI [19.06; 30.62]), 48.38% at 4 to 6 months (95% CI [41.68; 55.13]), 59.14% at 7 to 9 months (95% CI [50.88; 66.91]), 48.41% at 10 to 12 months (95% CI [41.54; 55.35]), 42.00% at 13 to 18 months (95% CI [37.01; 47.16]), and 48.34% at 19 to 24 months (95% CI [38.50; 58.31]). The highest NSPC rates were observed among children aged 4 to 6 months and 7 to 9 months across all vaccine groups. Low-income countries consistently demonstrated the highest NSPC rates across all age categories studied. This systematic review and meta-analysis provide robust evidence of the high prevalence of NSPC in infants aged 4 to 6 months and 7 to 9 months in all vaccine groups, with persistent regional disparities, especially among low-income countries. The study highlights the need for continuous monitoring of NSPC trends, particularly the emergence of non-vaccine serotypes. Policymakers and healthcare providers should leverage these findings to enhance vaccination strategies, aiming to minimize the overall burden of pneumococcal diseases in infants.

**Data Availability Statement:** All relevant data are within the manuscript and its Supporting Information files.

**Funding:** The author(s) received no specific funding for this work.

## Introduction

Streptococcus *pneumoniae* (SP) is a major global pathogen responsible for various invasive diseases, including pneumonia, meningitis, and septicemia, particularly affecting infants and young children [1]. The burden of pneumococcal disease is most acute in developing countries, where under-five mortality rates are significantly higher due to limited access to healthcare and preventive measures [2, 3]. SP not only leads to significant morbidity and mortality but also poses substantial economic and public health challenges worldwide [4]. Despite significant advancements in vaccination, which have dramatically reduced the incidence of pneumococcal disease, SP remains a critical public health concern [5, 6].

The introduction of pneumococcal vaccines has markedly reduced the incidence of pneumococcal diseases. For instance, a study from the United States showed that the mean rates of invasive pneumococcal disease (IPD) decreased by 40% among infants aged 0 to 90 days after the introduction of 7-valent Pneumococcal Conjugate Vaccine (PCV7) [7]. Similarly, a ten-year surveillance study from Gambia demonstrated that, in the 0 to 11 months age group, the introduction of 13-valent PCV (PCV13) eliminated vaccine-type IPD, with overall IPD incidence declining from 184 to 38 cases per 100,000 person-years [8]. However, a study from France indicated that while PCV7 introduction decreased rates of meningitis and bacteremia from vaccine strains, it increased rates of these diseases due to non-vaccine strains among children under 2 years of age [9]. Thus, studies suggest that the efficacy of PCV introduction varies significantly across different settings.

Examining the nasopharyngeal Streptococcus *pneumoniae* carriage (NSPC) rate in infants provides valuable insights into the epidemiology of pneumococcal disease [10, 11]. The carriage rate is a crucial indicator of the potential for transmission and the effectiveness of vaccination programs. Studies have shown that even in vaccinated populations, the carriage of non-vaccine serotypes can persist, highlighting the need for ongoing surveillance and vaccine updates [12]. Furthermore, a study in Malawi showed that 3 to 7 years after PCV13 introduction, there is a high residual vaccine-type SP carriage among children [13]. A meta-analysis of the invasive disease potential of SP showed that some of the non-vaccine serotypes 8, 12F, 24F, and 33F have high invasiveness potential among children under five years of age [14].

Thus, comparing carriage rates based on vaccination status and country income levels can inform public health strategies to mitigate the burden of pneumococcal disease and ensure that vaccination programs are achieving their intended outcomes. This systematic review and meta-analysis aim to provide a comprehensive overview of the prevalence of NSPC in infants during the first two years of their life, stratified by vaccine type and country income levels. By synthesizing data from cohort studies and randomized controlled trials, this study seeks to assess the impact of vaccination and major socio-economic factors, such as country income level, on pneumococcal carriage.

## Materials and methods

The study protocol is registered with the PROSPERO International prospective register of systematic reviews [15] (ID: CRD42024564709). The Local Ethics Committee of the Kazakh National Medical University named after S.D. Asfendiyarov approved this study (Study ID: 1276, date: 22.12.2021).

### Search strategy

The PROSPERO database was searched to identify registrations of comparable studies, but no similar study protocols were found. Subsequently, a comprehensive search was conducted in four electronic literature databases: PubMed, Web of Science, ScienceDirect, and Scopus from

April 1, 2024, to July 1, 2024. The search was structured according to the Population, Intervention, Comparator, Outcomes, and Study Design (PICOS) framework as follows: Population (P): infants and children under two years of age; Intervention (I): not applicable; Comparator (C): not applicable; Outcomes (O): nasopharyngeal carriage; and Study Design (S): cohort studies and randomized clinical trials (RCTs). The search strategy incorporated the following keywords: "Streptococcus *pneumoniae*" OR "pneumococcal" AND "carriage" AND "children under 2 years" OR "infants." No restrictions were imposed on the publication dates of studies. The complete search strategy, including additional restrictions applied to each database, is presented in S1 Table.

## Eligibility criteria

Reference screening and synthesis were conducted according to the Preferred Reporting Items for Systematic Reviews and Meta-Analyses (PRISMA) guidelines [16].

Inclusion criteria were:

1. Longitudinal cohort or randomized clinical trial studies reporting NSPC rates in healthy children under 2 years of age in peer-reviewed publications.

2. Studies conducted before and after the implementation of PCVs, including PCV7, 9-valent PCV (PCV9), 10-valent PCV (PCV10), 11-valent PCV (PCV11), PCV13, 23-valent PCV (PCV23).

3. Repetitive nasopharyngeal swabs conducted to collect samples for detecting Streptococcus *Pneumonia*e and pneumococcal carriage, analyzed using standard bacteriological culture, isolation, and identification methods.

4. Studies reporting longitudinal data with NSPC for specific age groups: 0 months, 1–3 months, 4–6 months, 7–9 months, 10–12 months, 13–18 months, and 18–24 months (including 18–27 months as an exception).

5. Publications in English.

Exclusion criteria were:

1. Studies not focusing on the specified age group.

2. Cross-sectional, case-control studies, and clinical trial protocols. Cross-sectional data within longitudinal cohort studies or RCTs were also excluded.

3. Studies focusing on unhealthy children, including but not limited to otitis media, acute respiratory infections, HIV-diagnosed infants, meningitis, and others.

4. Studies reporting longitudinal data for different age groups, such as 7–12 months, 20–30 months, etc.

5. Articles assessing specific SP serotype distribution or immunoglobulin levels.

## Selection of studies and data extraction

Two authors independently screened the titles and abstracts of the search results for relevance after removing duplicates. The full texts of the eligible studies were then assessed against the inclusion and exclusion criteria. The following data were extracted: study characteristics (author, year, location, study design, and study period); population details (age in months and sample size); repetitive measurement age; vaccination status; vaccine name and schedule if

vaccinated; longitudinal SP carriage rates for specified age groups; and serotyping method. Discrepancies were resolved through consultation with a third author.

### Risk of bias and study quality assessment

The risk of bias and quality of the included studies were assessed using the Critical Appraisal Skills Programme (CASP) Checklist for Cohort Studies and CASP Checklist for Randomized Controlled Trials (RCTs) [17]. The CASP checklist for cohort studies includes 12 questions assessing the validity of the results, the precise measurement of outcomes, and the consideration of confounding factors. The CASP checklist for RCTs also contains 12 questions evaluating randomization, blinding, the handling of withdrawals and dropouts, and the assessment of results. In both questionnaires the questions are answered with "yes," "no," or "can't tell," with "yes" scoring 1, "no" scoring 0, and "can't tell" scoring 0.5. The official guidelines for the use of the CASP checklist do not provide recommendations for the interpretation of the scores. Therefore, the authors uniformly decided to include articles that scored 7 or above on both questionnaires.

### Statistical analysis

Subgroup analysis based on vaccination status and country income level was used to calculate the pooled mean prevalence of NSPC for each specified age group, along with 95% confidence intervals (95% CI), using a random-effects model for meta-analysis in RStudio software with the "meta" and "metafor" packages [18]. Heterogeneity across studies was assessed using the $I^2$-statistic. The Egger test was computed to evaluate publication bias. As a final step of the analysis, data on NSPC among different age groups, based on vaccine type, were combined and visualized using a bar graph with 95% confidence intervals to compare NSPC rates among different vaccine types.

## Results

A comprehensive search across PubMed, Web of Science, ScienceDirect, and Scopus electronic databases resulted in 1,083 records. After initial screening, 309 duplicate records were removed, and 774 references were assessed for relevance. Of these, 442 full-text articles were sought for retrieval, one article was not assessed, and ultimately, 49 articles met the criteria for inclusion in the systematic review and meta-analysis. The study selection process is illustrated in the PRISMA flow diagram in Fig 1 [16]. Turner et al. [19], and Coles et al. [20] presented the same data as already included articles and were therefore excluded. The complete study selection process results are presented in S2 Table.

### Methods of studies

A total of 49 articles published between 1996 and 2023 from 23 countries met the inclusion criteria for this analysis. Seven articles originated from Gambia, four from India, and four from Papua New Guinea. Three articles each were from Finland, Israel, the Netherlands, South Africa, and the United States of America (USA). Two articles each came from Australia, Brazil, Indonesia, and the Philippines. There was one study from each of the following countries: Ethiopia, Germany, India and Bangladesh, Japan, Jordan, Kenya, Malawi, Taiwan, Thailand and Myanmar, the United Kingdom, and Vietnam.

Three articles did not provide information on the serotyping method. Among the 46 articles that did, 27 used the Quellung reaction for serotyping, 13 used the PCR method, and the

**PRISMA 2020 flow diagram for new systematic reviews which included searches of databases and registers only**

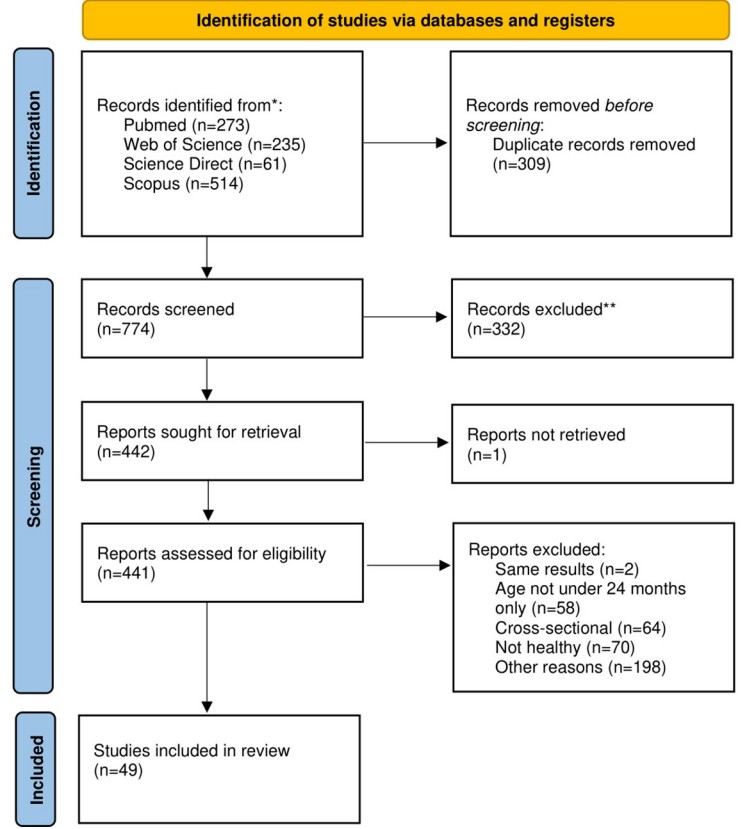

**Fig 1. PRISMA flow diagram of study selection [16].**

rest used different methods. Descriptions of the included studies are presented in Table 1. The complete data extraction table is presented in S3 Table.

## The risk of bias and study quality assessment

The risk of bias and study quality assessment results are presented in Table 2. Twenty-six cohort studies were evaluated using the CASP checklist for cohort studies, and twenty-three RCT's and vaccine trials were evaluated using the CASP checklist for RCTs. All of the studies had a CASP score 7 and above.

## Nasopharyngeal Streptococcus *Pneumoniae* carriage

In this section, we present the prevalence of nasopharyngeal SP carriage in infants during the first two years of life, stratified by vaccine type and country income level. The data are organized by age intervals as follows: 0 months, 1–3 months, 4–6 months, 7–9 months, 10–12 months, 13–18 months, and 19–24 months.

Eight studies with nine groups presented the data on the NSPC at birth. According to the results obtained from the random effects model, the pooled mean prevalence of NSPC at birth was 1.68%, 95% CI [0.50; 5.47], with high heterogeneity. Subgroup analysis by vaccine type showed NSPC rates as follows: in the no-vaccine group, 1.73% (95% CI [0.79; 3.75]); in the

**Table 1. Description of the included articles.**

| # | author | study design | country | when data was collected | serotyping |
|---|---|---|---|---|---|
| 1 | Dagan, 1996 [21] | longitudinal cohort | Israel | November 1993 through March 1994 in a comparative conjugate Rib vaccine study. | Quellung reaction |
| 2 | Obaro, 2000 [22] | vaccine trial | Gambia | no exact dates are given | type- or group-specific capsular polysaccharide antibody |
| 3 | Coles, 2001 [23] | longitudinal cohort | India | 1998–2001 | PNEUMOTEST kits |
| 4 | Leino, 2001 [24] | longitudinal cohort | Finland | 1994–1997 | Counterimmunoelectrophoresis and latex agglutination with the use of antiserum pools and group/type specific antisera |
| 5 | Syrjänen, 2001 [25] | longitudinal cohort | Finland | April 1994 and August 1995 | Counterimmunoelectrophoresis and latex agglutination, using antiserum pools and group- and type-specific antisera |
| 6 | Yeh, 2003 [26] | longitudinal cohort | USA | June–October 1995 | Quellung reaction |
| 7 | Ghaffar, 2004 [27] | longitudinal cohort | USA | September 2000 through August 2001. | Neufeld-Quellung reaction |
| 8 | Holmlund, 2006 [28] | vaccine trial | Philippines | July 1994 to September 1995 | no info |
| 9 | Darboe, 2007 [29] | RCT | Gambia | September, 2001, and October, 2004 | no info |
| 10 | Hill, 2008 [30] | longitudinal cohort | Gambia | No dates are provided | Quellung reaction |
| 11 | Labout, 2008 [31] | longitudinal cohort | Netherlands | June 2003 and November 2006. | Quellung reaction |
| 12 | Cheung, 2009 [32] | vaccine trial | Gambia | until April 30, 2004 | Quellung reaction |
| 13 | Lopes, 2009 [33] | RCT of women | Brazil | May 2005 to January 2006. | no info |
| 14 | van Gils, 2009 [34] | vaccine trial | Netherlands | July 7, 2005, and February 14, 2008 | Quellung reaction |
| 15 | Darboe, 2010 [35] | longitudinal cohort | Gambia | in the preparation for the introduction and evaluation of the 13-valent conjugate | Quellung reaction |
| 16 | Väkeväinen, 2010 [36] | vaccine trial | Philippines | 2000–2004 | Chessboard modification of the quellung method |
| 17 | Kwambana, 2011 [37] | longitudinal cohort | Gambia | No dates are provided | PCR |
| 18 | Scott, 2011 [38] | vaccine trial | Kenya | no exact dates are given. The KHDSS was used to follow long-term mortality up to July 2009. | Quellung reaction |
| 19 | Coles, 2012 [39] | RCT | India | October 1998 through June 1999 | PNEUMOTEST kits (Staten Serum Institute) |
| 20 | Dagan, 2012 [40] | vaccine trial | Israel | This open-label study was initiated in August 2005. Enrollment lasted through March 2008, and last follow-up visit was in March 2009 | Quellung reaction |
| 21 | Lopes, 2012 [41] | longitudinal cohort | Brazil | May 2005 and January 2006. | Quellung reaction |
| 22 | Turner, 2012 [42] | longitudinal cohort | Thailand-Myanmar | October 2007 and November 2008 | Latex agglutination with Quellung confirmation |
| 23 | Otsuka, 2013 [43] | longitudinal cohort | Japan | January 2009—December 2011 | Quellung reaction |
| 24 | Biesbroek, 2014 [44] | longitudinal cohort | Netherlands | The study was initiated shortly before nationwide implementation of PCV7 in the Netherlands. | no info |
| 25 | Rupa, 2014 [45] | longitudinal cohort | India | February and August | Coagglutination technique with antisera obtained from Statens Seruminstitut |

*(Continued)*

**Table 1.** (Continued)

| # | author | study design | country | when data was collected | serotyping |
|---|--------|--------------|---------|------------------------|------------|
| 26 | Binks, 2015 [46] | RCT of women | Australia | August 2006 and January 2011 | Quellung reaction |
| 27 | Aho, 2016 [47] | vaccine trial | Papua New Guinea | 2005 and 2009 | Quellung reaction |
| 28 | Vesikari, 2016 [48] | RCT | Finland | February 2009 and December 2011 | Quellung reaction |
| 29 | Al-Lahham, 2018 [49] | vaccine trial | Jordan | March and April of 2009 | Neufeld's Quellung reaction |
| 30 | Dagan, 2018 [50] | vaccine trial | Israel | August 2005. Enrollment was conducted through March 2008, and the last follow-up visit was in March 2009 | Quellung reaction |
| 31 | Dube, 2018 [51] | longitudinal cohort | South Africa | May 29th 2012 and May 31st 2014 | lytA PCR |
| 32 | Usuf, 2018 [52] | longitudinal cohort | Gambia | April 2013 and April 2014 | Latex agglutination test |
| 33 | Murad, 2019 [53] | longitudinal cohort | Indonesia | November 2014 to January 2015 | lytA real-time quantitative PCR (qPCR) |
| 34 | Pomat, 2019 [54] | vaccine trial | Papua New Guinea | November 2011 and April 2014 | Quellung reaction |
| 35 | Sime, 2019 [55] | longitudinal cohort | Ethiopia | February 2013 to November 2016 | Quellung reactions |
| 36 | Vanker, 2019 [56] | longitudinal cohort | South Africa | March 2012 to July 2015 | lytA PCR |
| 37 | Meropol, 2020 [57] | longitudinal cohort | USA | April 2013–February 2014 | According to the Clinical and Laboratory Standards Institute Standards for Antimicrobial Susceptibility Testing |
| 38 | Nunes, 2020 [58] | vaccine trial | South Africa | Dec2009-Apr2010 and Mar2009-May2010 in the PCV7 and PHiD-CV studies | Quellung reaction |
| 39 | Tsai, 2020 [59] | longitudinal cohort | Taiwan | launched in 2013 | Antisera (Statens Serum Institut, Copenhagen, Denmark) and polymerase chain reaction (PCR) methods |
| 40 | Apte, 2021 [60] | longitudinal cohort | India and Bangladesh | December 2016 to May 2018 | Quellung test and polymerase chain reaction (PCR) |
| 41 | Britton, 2021 [61] | RCT | Papua New Guinea | 2011–2016 and 2013–2016 | qPCR |
| 42 | Prayitno, 2021 [62] | longitudinal cohort | Indonesia | from March 2018 until June 2019 | conventional multiplex PCR |
| 43 | Rose, 2021 [63] | longitudinal cohort | Germany | October 2008 and June 2009 | Neufeld Quellung reaction |
| 44 | Dherani, 2022 [64] | longitudinal cohort | Malawi | Nov 15, 2015, and Nov 2, 2017 | PCR |
| 45 | Martinovich, 2022 [65] | RCT | Australia | 2001–2009 | PCR |
| 46 | Goldblatt, 2023 [66] | vaccine trial | United Kingdom | In January 2020 the UK changed from a 2 + 1 schedule for 13-valent pneumococcal conjugate vaccine (PCV13) to a 1 + 1 schedule | Serological analysis was performed at the World Health Organisation (WHO) reference laboratory for pneumococcal serology, Great Ormond Street Institute of Child Health, University College London. |
| 47 | Kawade, 2023 [67] | vaccine trial | India | July 2016 to May 2018 | Quellung reaction |
| 48 | Orami, 2023 [68] | vaccine trial | Papua New Guinea | November 2011 and March 2016 | Quellung reaction |
| 49 | Temple, 2023 [69] | vaccine trial | Vietnam | March 8, 2017, and July 24, 2018 | Quellung reaction |

**Table 2. CASP risk of bias and study quality assessment.**

**CASP checklist for Cohort studies**

| Author, year | Aim | Methodology | Design | Recruitment | Data collection | Relationship | Ethical | Data analysis | Finding | Values | Score |
|---|---|---|---|---|---|---|---|---|---|---|---|
| Dagan, 1996 [21] | Yes | Yes | No | Can't tell | Yes | No | Yes | Yes | Yes | Yes | 7.5 |
| Coles, 2001 [23] | Yes | Yes | No | Can't tell | Yes | No | Yes | Yes | Yes | Yes | 7.5 |
| Leino, 2001 [24] | Yes | Yes | No | Can't tell | Yes | No | Yes | Yes | Yes | Yes | 7.5 |
| Syrjänen, 2001 [25] | Yes | Yes | No | Can't tell | Yes | No | Yes | Yes | Yes | Yes | 7.5 |
| Yeh, 2003 [26] | Yes | Yes | No | Can't tell | Yes | No | Yes | Yes | Yes | Yes | 7.5 |
| Ghaffar, 2004 [27] | Yes | Yes | No | Can't tell | Yes | No | Yes | Yes | Yes | Yes | 7.5 |
| Hill, 2008 [30] | Yes | Can't tell | No | Can't tell | Yes | Yes | Yes | Yes | Yes | Yes | 7 |
| Labout, 2008 [31] | Yes | Yes | No | Can't tell | Yes | No | Yes | Yes | Yes | Yes | 7.5 |
| Darboe, 2010 [35] | Yes | Yes | No | Can't tell | Yes | No | Yes | Yes | Yes | Yes | 7.5 |
| Kwambana, 2011 [37] | Yes | Yes | No | Can't tell | Yes | No | Yes | Yes | Yes | Yes | 7.5 |
| Lopes, 2012 [41] | Yes | Yes | No | Can't tell | Yes | No | Yes | Yes | Yes | Yes | 7.5 |
| Turner, 2012 [42] | Yes | Yes | No | Can't tell | Yes | No | Yes | Yes | Yes | Yes | 7.5 |
| Otsuka, 2013 [43] | Yes | Can't tell | No | Can't tell | Yes | Yes | Yes | Yes | Yes | Yes | 7 |
| Biesbroek, 2014 [44] | Yes | Yes | No | Can't tell | Yes | No | Yes | Yes | Yes | Yes | 7.5 |
| Rupa, 2014 [45] | Yes | Yes | No | Can't tell | Yes | No | Yes | Yes | Yes | Yes | 7.5 |
| Dube, 2018 [51] | Yes | Yes | No | Can't tell | Yes | No | Yes | Yes | Yes | Yes | 7.5 |
| Usuf, 2018 [52] | Yes | Yes | No | Can't tell | Yes | No | Yes | Yes | Yes | Yes | 7.5 |
| Murad, 2019 [53] | Yes | Yes | No | Can't tell | Yes | No | Yes | Yes | Yes | Yes | 7.5 |
| Sime, 2019 [55] | Yes | Can't tell | No | Can't tell | Yes | Yes | Yes | Yes | Yes | Yes | 7 |
| Vanker, 2019 [56] | Yes | Yes | No | Can't tell | Yes | No | Yes | Yes | Yes | Yes | 7.5 |
| Meropol, 2020 [57] | Yes | Yes | No | Can't tell | Yes | No | Yes | Yes | Yes | Yes | 7.5 |
| Tsai, 2020 [59] | Yes | Yes | No | Can't tell | Yes | No | Yes | Yes | Yes | Yes | 7.5 |
| Apte, 2021 [60] | Yes | Yes | No | Can't tell | Yes | No | Yes | Yes | Yes | Yes | 7.5 |
| Prayitno, 2021 [62] | Yes | Yes | No | Can't tell | Yes | No | Yes | Yes | Yes | Yes | 7.5 |
| Rose, 2021 [63] | Yes | Can't tell | No | Can't tell | Yes | Yes | Yes | Yes | Yes | Yes | 7 |
| Dherani, 2022 [64] | Yes | Can't tell | No | Can't tell | Yes | Yes | Yes | Yes | Yes | Yes | |

**CASP checklist for RCT**

| Author, year | Aim | Methodology | Design | Recruitment | Data collection | Relationship | Ethical | Data analysis | Finding | Values | Score |
|---|---|---|---|---|---|---|---|---|---|---|---|
| Obaro, 2000 [22] | Yes | Yes | No | Can't tell | Yes | No | Yes | Yes | Yes | Yes | 7.5 |
| Holmlund, 2006 [28] | Yes | Yes | No | Can't tell | Yes | No | Yes | Yes | Yes | Yes | 7.5 |
| Darboe, 2007 [29] | Yes | Yes | No | Can't tell | Yes | No | Yes | Yes | Yes | Yes | 7.5 |
| Cheung, 2009 [32] | Yes | Yes | No | Can't tell | Yes | No | Yes | Yes | Yes | Yes | 7.5 |
| Lopes, 2009 [33] | Yes | Yes | No | Can't tell | Yes | No | Yes | Yes | Yes | Yes | 7.5 |
| van Gils, 2009 [34] | Yes | Can't tell | No | Can't tell | Yes | Yes | Yes | Yes | Yes | Yes | 7 |
| Vakevainen, 2010 [36] | Yes | Yes | No | Can't tell | Yes | No | Yes | Yes | Yes | Yes | 7.5 |
| Scott, 2011 [38] | Yes | Yes | No | Can't tell | Yes | No | Yes | Yes | Yes | Yes | 7.5 |
| Coles, 2012 [39] | Yes | Yes | No | Can't tell | Yes | No | Yes | Yes | Yes | Yes | 7.5 |
| Dagan, 2012 [40] | Yes | Yes | No | Can't tell | Yes | No | Yes | Yes | Yes | Yes | 7.5 |
| Binks, 2015 [46] | Yes | Yes | No | Can't tell | Yes | No | Yes | Yes | Yes | Yes | 7.5 |
| Aho, 2016 [47] | Yes | Can't tell | No | Can't tell | Yes | Yes | Yes | Yes | Yes | Yes | 7 |
| Vesikari, 2016 [48] | Yes | Yes | No | Can't tell | Yes | No | Yes | Yes | Yes | Yes | 7.5 |
| Al-Lahham, 2018 [49] | Yes | Yes | No | Can't tell | Yes | No | Yes | Yes | Yes | Yes | 7.5 |
| Dagan, 2018 [50] | Yes | Yes | No | Can't tell | Yes | No | Yes | Yes | Yes | Yes | 7.5 |
| Pomat, 2019 [54] | Yes | Yes | No | Can't tell | Yes | No | Yes | Yes | Yes | Yes | 7.5 |
| Nunes, 2020 [58] | Yes | Yes | No | Can't tell | Yes | No | Yes | Yes | Yes | Yes | 7.5 |
| Britton, 2021 [61] | Yes | Can't tell | No | Can't tell | Yes | Yes | Yes | Yes | Yes | Yes | 7 |

*(Continued)*

**Table 2.** (Continued)

| | | | | | | | | | | | |
|---|---|---|---|---|---|---|---|---|---|---|---|
| Martinovich, 2022 [65] | Yes | Yes | No | Can't tell | Yes | No | Yes | Yes | Yes | Yes | 7.5 |
| Goldblatt, 2023 [66] | Yes | Yes | No | Can't tell | Yes | No | Yes | Yes | Yes | Yes | 7.5 |
| Kawade, 2023 [67] | Yes | Yes | No | Can't tell | Yes | No | Yes | Yes | Yes | Yes | 7.5 |
| Orami, 2023 [68] | Yes | Yes | No | Can't tell | Yes | No | Yes | Yes | Yes | Yes | 7.5 |
| Temple, 2023 [69] | Yes | Yes | No | Can't tell | Yes | No | Yes | Yes | Yes | Yes | 7.5 |

Abbreviations: CASP: Critical Appraisal Skills Programme

PCV7 group, 21.56% (95% CI [17.05; 26.88]); in the PCV7 and PCV13 group, 1.60% (95% CI [0.06; 29.59]); and in the PCV13 group, 0% (95% CI [0; 0.87]) (Fig 2).

Subgroup analysis by income level showed NSPC rates as follows: in the high income countries group, 1.71%, 95% CI [0.08; 28.49]; in the upper-middle income countries group, 0%, 95% CI [0–0.87]; in the lower middle income countries group, 9.69%, 95% CI [1.58; 41.72]; in the low income countries group, 1.22%, 95% CI [0.58; 2.55] (Fig 3).

Thirty-three articles with fifty-five groups presented data on the NSPC rate at 1 to 3 months of age. The random-effects model results indicated a pooled mean prevalence of 24.38% (95% CI: 19.06; 30.62), with substantial heterogeneity. Subgroup analysis by vaccine type showed NSPC rates as follows: in the no-vaccine group, 24.89%, 95% CI [19.06; 30.62]; in the PCV7 group, 21.75%, 95% CI [14.58; 31.17]; in the PCV10 group, 14.88%, 95% CI [6.19; 31.63]; in

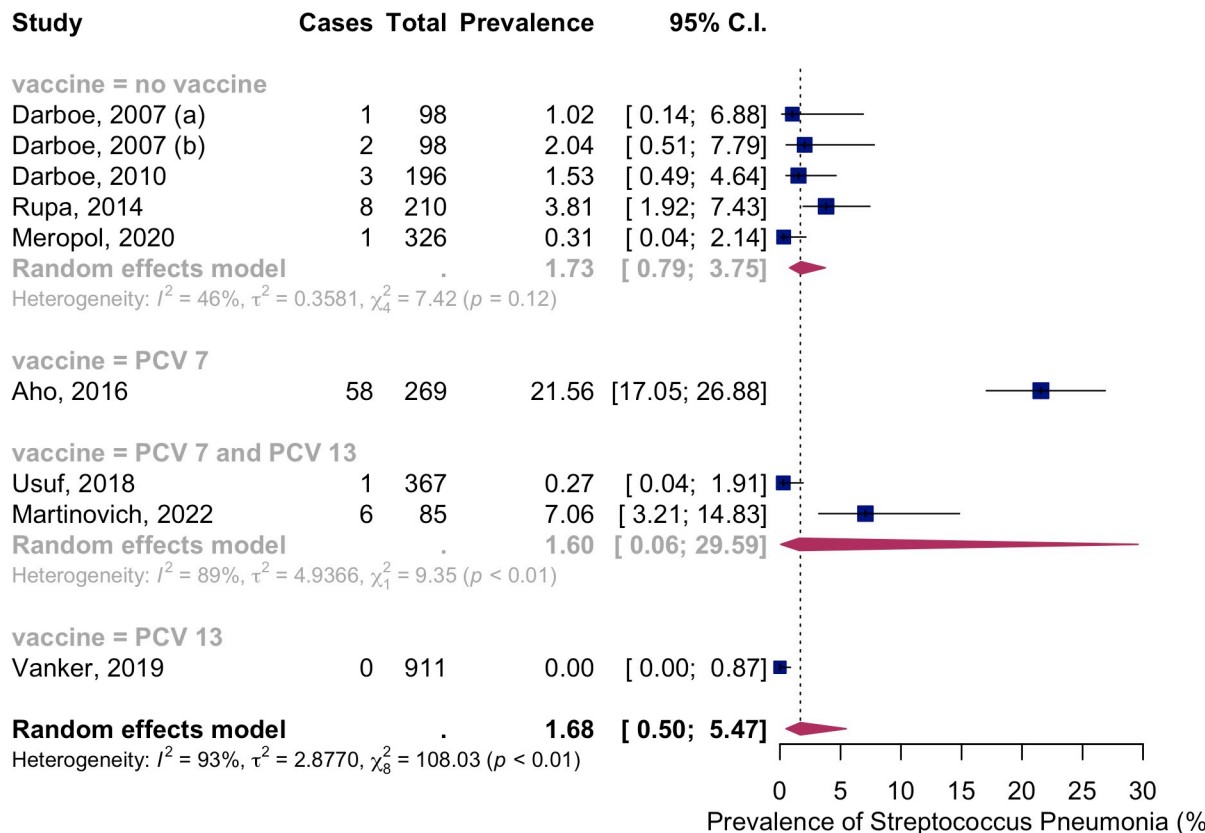

**Fig 2. Prevalence of the nasopharyngeal Streptococcus *Pneumoniae* carriage among infants at 0 months: Subgroup analysis based on vaccine type.** Abbreviations: C.I.: confidence interval.

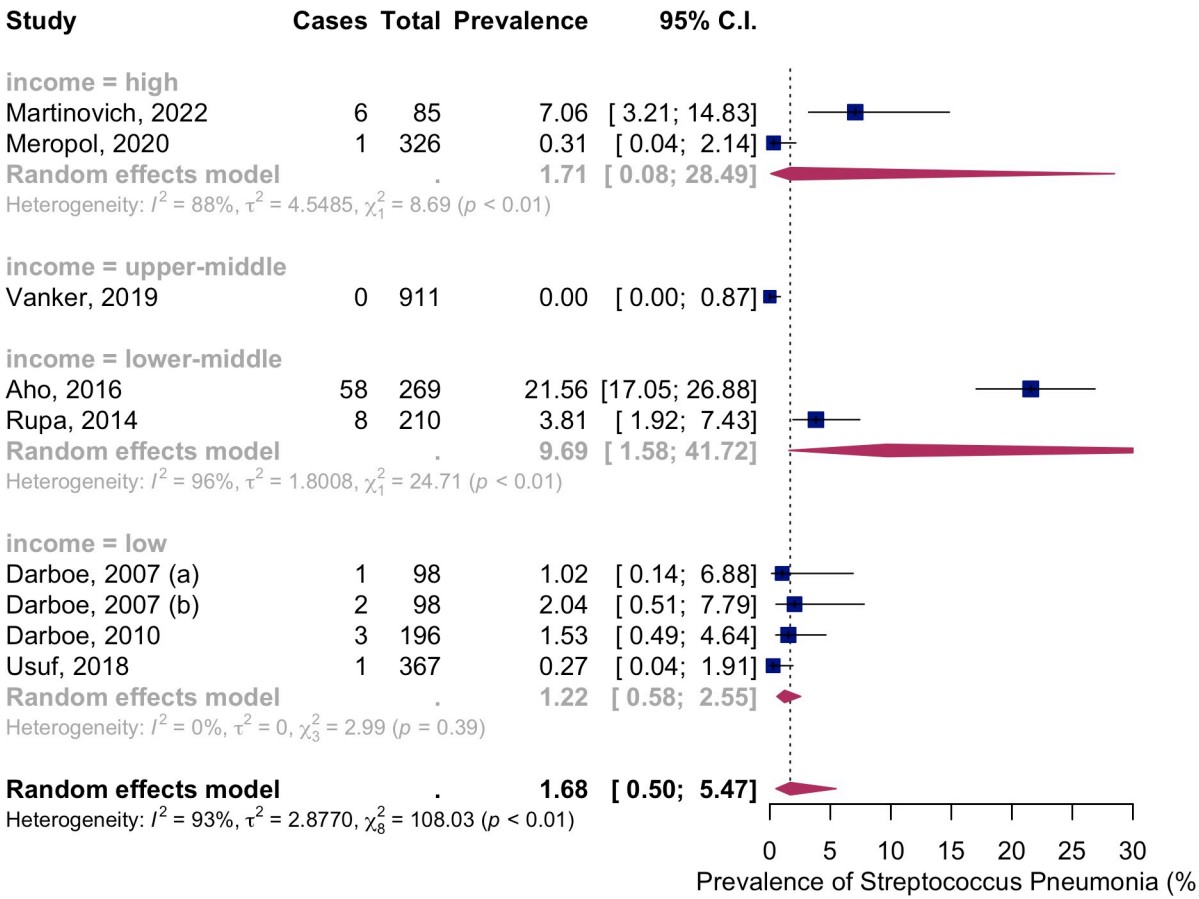

**Fig 3. Prevalence of the nasopharyngeal Streptococcus *Pneumoniae* carriage among infants at 0 months: Subgroup analysis based on country income level.** Abbreviations: C.I.: confidence interval.

the PCV13 group, 39.17%, 95% CI [26.30; 53.76]; in the PCV23 group, 28.32%, 95% CI [15.71; 45.58]; in the mixed group, 22.11%, 95% CI [9.71; 42.84] (Fig 4).

Subgroup analysis by income level showed NSPC rates as follows: in the high income countries group, 11.87%, 95% CI [7.53; 18.22]; in the upper-middle income countries group, 25.99%, 95% CI [16.97; 37.64]; in the lower middle income countries group, 30.64%, 95% CI [21.56; 41.51]; in the low income countries group, 55.42%, 95% CI [38.25; 71.38] (Fig 5).

Thirty-two articles with fifty-five groups presented data on the NSPC rate at 4 to 6 months of age. According to the random effects model results, the pooled mean prevalence of NSPC at 4 to 6 months of age was 48.38%, 95% CI [41.68; 55.13]< with high heterogeneity. Subgroup analysis by vaccine type showed NSPC rates as follows: in the no vaccine group, 50.15%, 95% CI [40.21; 60.08]; in the PCV7 group, 48.14%, 95% CI [20.64; 76.81]; in the PCV9 group, 92.00%, 95% CI [84.81; 95.95]; in the PCV10 group, 46.86%, 95% CI [33.40; 60.79]; in the PCV11 group, 28.57%, 95% CI [24.89; 32.56]; in the PCV13 group, 55.48%, 95% CI [37.18; 72.41]; in the mixed group, 11.49%, 95% CI [4.63; 27.78]; in the PCV23 group, 34.28%, 95% CI [15.99; 58.83] (Fig 6).

Subgroup analysis by income level showed NSPC rates as follows: in the high income countries group, 21.26%, 95% CI [16.11; 27.52]; in the upper-middle income countries group, 35.41%, 95% CI [21.03; 53.03]; in the lower middle income countries group, 51.41%, 95% CI [42.20; 60.52]; in the low income countries group, 87.00%, 95% CI [84.90; 88.84] (Fig 7).

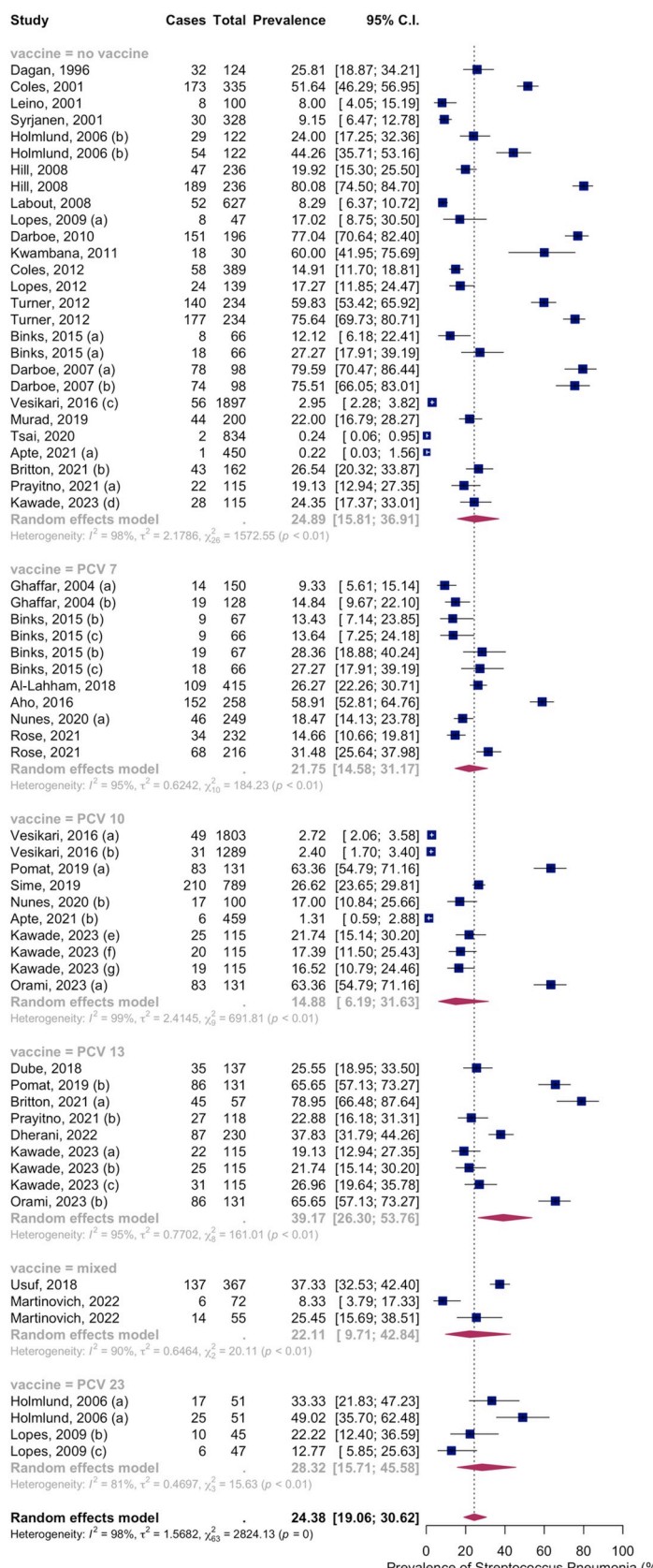

**Fig 4. Prevalence of the nasopharyngeal Streptococcus *Pneumoniae* carriage among infants at 1 to 3 months: Subgroup analysis based on vaccine type.** Abbreviations: C.I.: confidence interval.

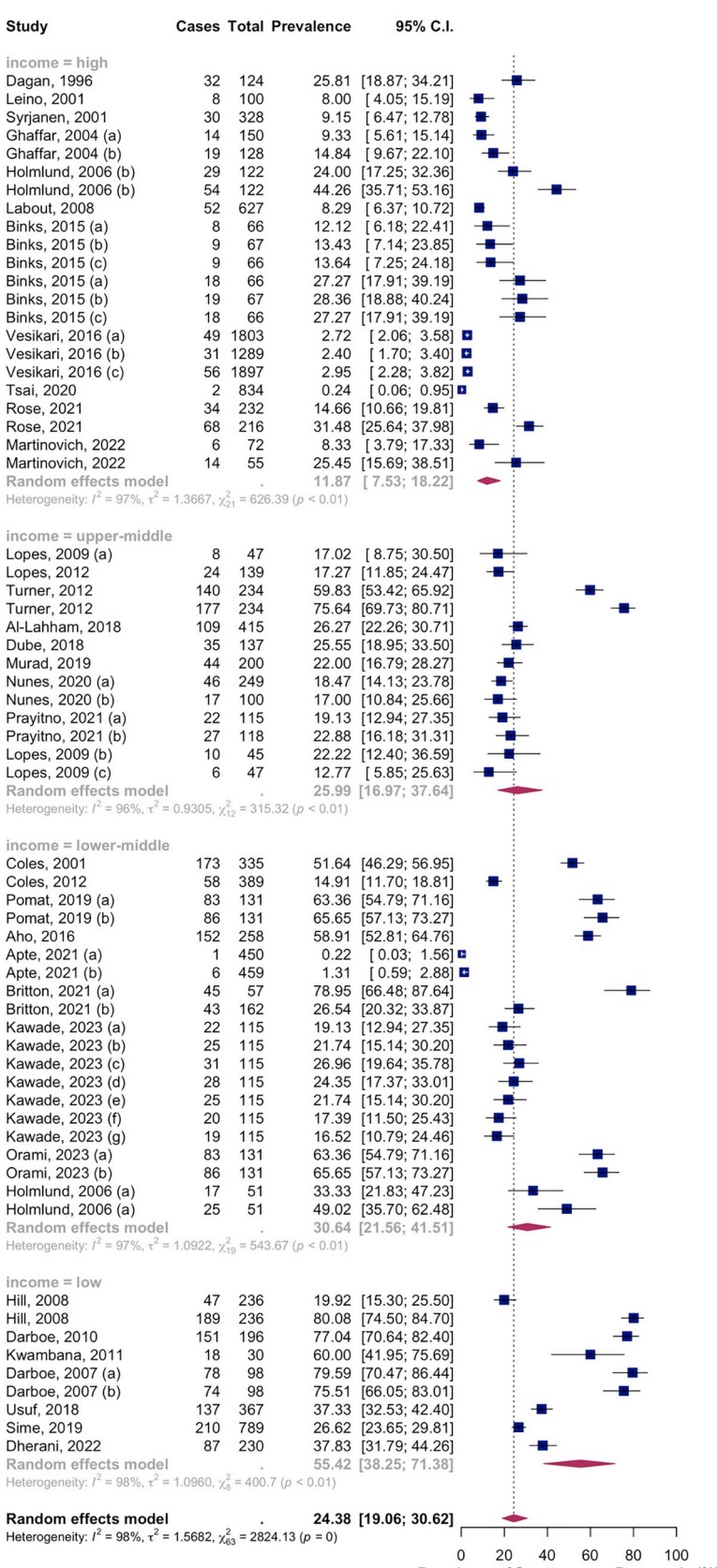

**Fig 5. Prevalence of the nasopharyngeal Streptococcus *Pneumoniae* carriage among infants at 1 to 3 months: Subgroup analysis based on country income level.** Abbreviations: C.I.: confidence interval.

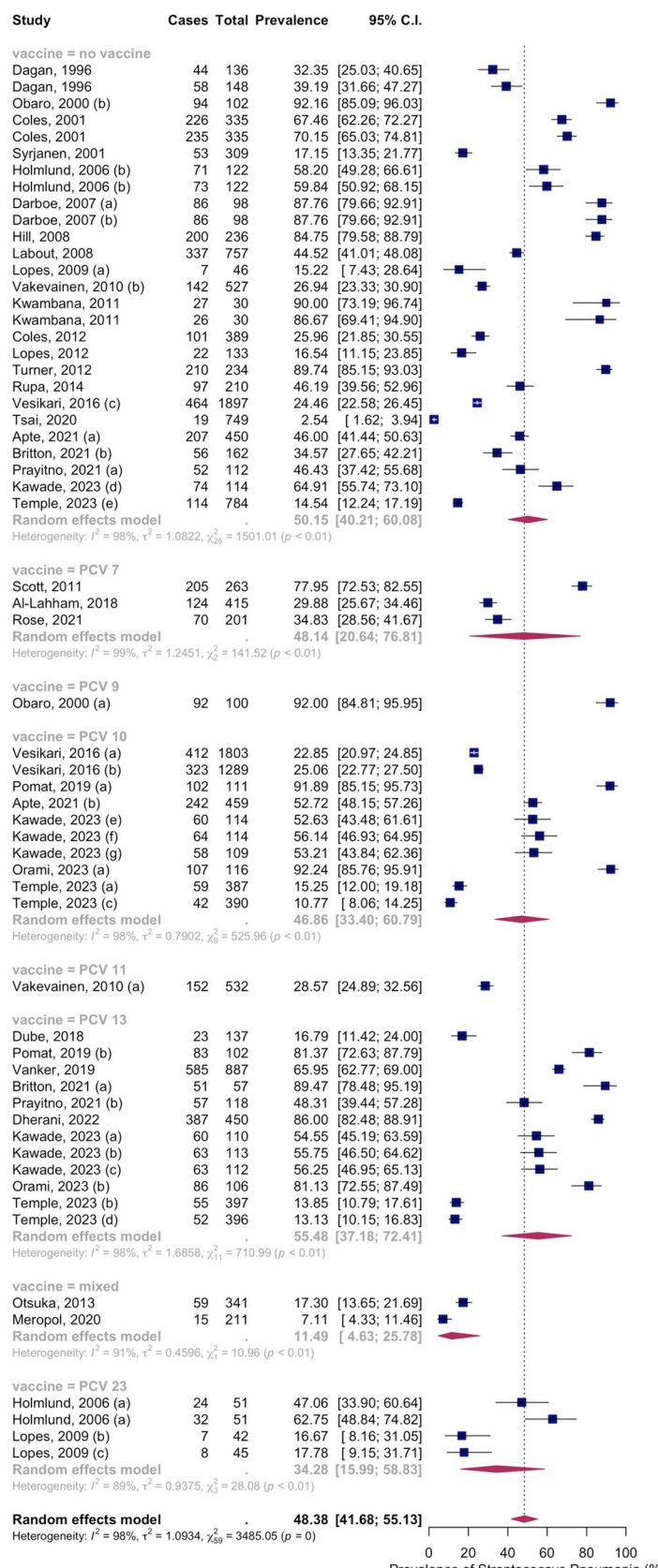

**Fig 6. Prevalence of the nasopharyngeal Streptococcus *Pneumoniae* carriage among infants at 4 to 6 months: Subgroup analysis based on vaccine type.** Abbreviations: C.I.: confidence interval.

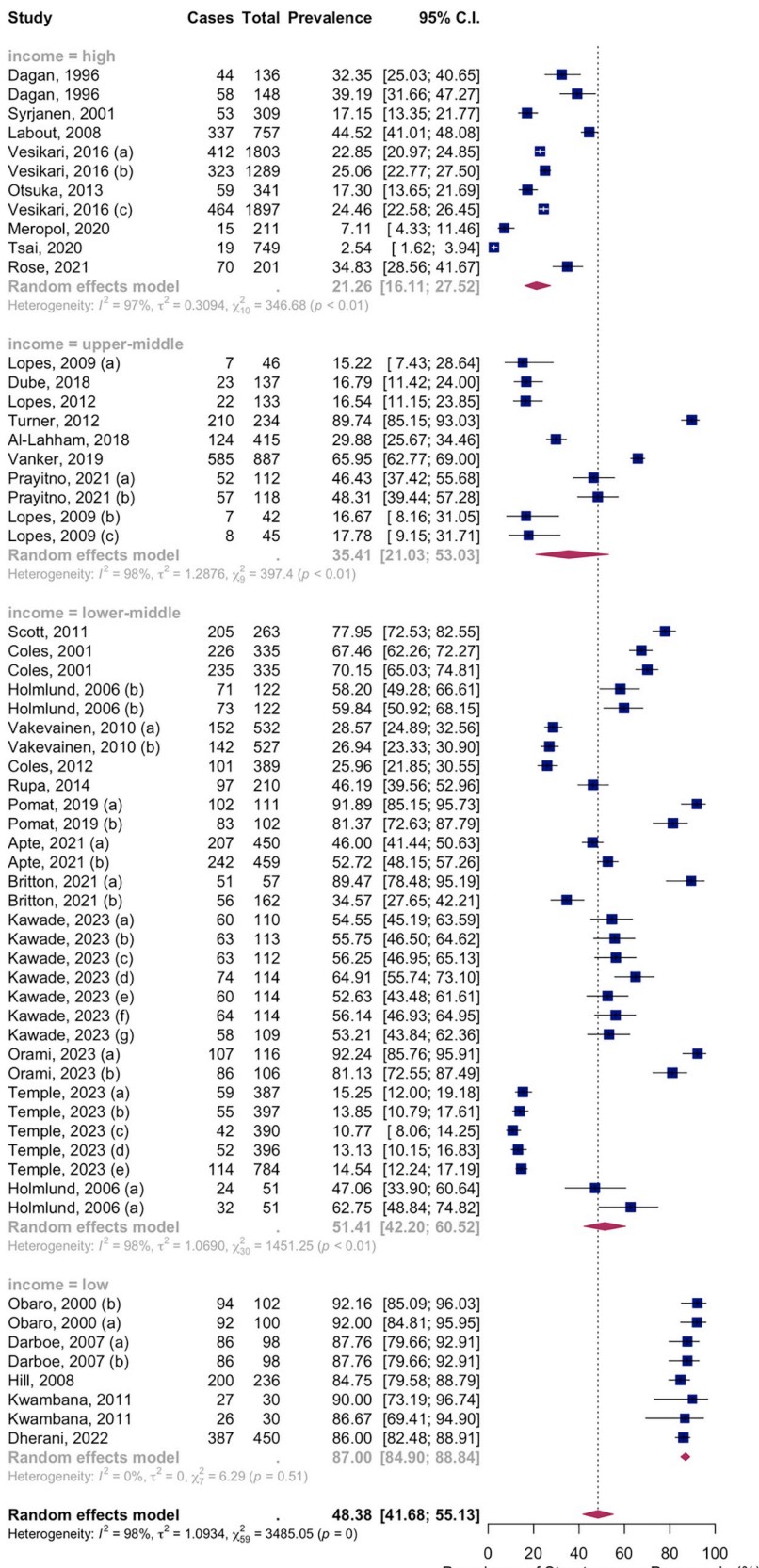

**Fig 7. Prevalence of the nasopharyngeal Streptococcus *Pneumoniae* carriage among infants at 4 to 6 months: Subgroup analysis based on country income level.** Abbreviations: C.I.: confidence interval.

Seventeen articles with thirty-three groups presented data on the NSPC rate at 7 to 9 months of age. According to the random effects model results, the pooled mean prevalence of NSPC at 7 to 9 months of age was 59.14%, 95% CI [50.88; 66.91], with high heterogeneity. Subgroup analysis by vaccine type showed NSPC rates as follows: in the no vaccine group, 47.62%, 95% CI [34.12; 61.48]; in the PCV7 group, 58.75%, 95% CI [38.34; 76.54]; in the PCV9 group, 84.69%, 95% CI [76.15; 90.56]; in the PCV10 group, 67.82%, 95% CI [58.47; 75.93]; in the PCV11 group, 28.38%, 95% CI [24.69; 32.39]; in the PCV13 group, 79.11%, 95% CI [65.32; 88.39]; in the mixed group, 20.56%, 95% CI [7.05; 46.90]; in the PCV23 group, 78.43%, 95% CI [65.11; 87.63] (Fig 8).

Subgroup analysis by income level showed NSPC rates as follows: in the high income countries group, 29.27%, 95% CI [18.24; 43.43]; in the lower middle income countries group, 67.70%, 95% CI [58.33; 75.83]; in the low income countries group, 78.58%, 95% CI [52.67; 92.36] (Fig 9).

Thirty-three articles with forty-five groups presented data on the NSPC rate at 10 to 12 months of age. According to the random effects model results, the pooled mean prevalence of NSPC at 10 to 12 months of age was 48.41%, 95% CI [41.54; 55.35], with high heterogeneity. Subgroup analysis by vaccine type showed NSPC rates as follows: in the no vaccine group, 48.54%, 95% CI [35.92; 61.36]; in the PCV7 group, 60.58%, 95% CI [49.96; 70.29]; in the PCV10 group, 45.45%, 95% CI [34.73; 56.62]; in the PCV13 group, 50.68%, CI [34.54; 66.69]; in the mixed group, 20.76%, 95% CI [5.25; 55.35]; in the PCV23 group, 58.82%, 95% CI [44.99; 71.39] (Fig 10).

Subgroup analysis by income level showed NSPC rates as follows: in the high income countries group, 37.00%, 95% CI [29.36; 45.35]; in the upper-middle income countries group, 56.58%, 95% CI [40.80; 71.13]; in the lower middle income countries group, 49.11%, 95% CI [36.45; 61.88]); in the low income countries group, 81.69%, 95% CI [75.92; 86.33] (Fig 11).

Sixteen articles with thirty-eight groups presented data on the NSPC rate at 13 to 18 months of age. According to the random effects model results, the pooled mean prevalence of NSPC at 13 to 18 months of age was 42.00%, 95% CI [37.01; 47.16], with high heterogeneity. Subgroup analysis by vaccine type showed NSPC rates as follows: in the no vaccine group, 38.44%, 95% CI [27.66; 50.48]; in the PCV7 group, 0.62%, 95% CI [32.03; 49.82]; in the PCV10 group, 39.10%, 95% CI [31.14; 47.69]; in the PCV13 group, 49.20%, 95% CI [35.32; 63.20]; in the mixed group, 48.00%, 95% CI [42.61; 53.43] (Fig 12).

Subgroup analysis by income level showed NSPC rates as follows: in the high-income countries group, 39.30%, 95% CI [33.20; 45.75]; in the upper-middle income countries group, 55.53%, 95% CI [26.43; 81.28]; in the lower middle income countries group, 43.95%, 95% CI [34.46; 53.91] (Fig 13).

Thirteen articles presented data on the NSPC rate at 19 to 24 months of age. According to the random effects model results, the pooled mean prevalence of NSPC at 19 to 24 months of age was 48.34%, 95% CI [38.50; 58.31], with high heterogeneity. Subgroup analysis by vaccine type showed NSPC rates as follows: in the no vaccine group, 47.03%, 95% CI [30.96; 63.74]; in the PCV7 group, 58.18%, 95% CI [54.10; 62.15]; in the PCV9 group, 82.01%, 95% CI [79.46; 84.30]; in the PCV10 group, 41.39%, 95% CI [30.47; 53.22]; in the PCV13 group, 37.72%, 95% CI [10.43; 75.91] (Fig 14).

Subgroup analysis by income level showed NSPC rates as follows: in the high income countries group, 43.91%, 95% CI [35.62; 52.54]; in the upper-middle income countries group, 66.97%, 95% CI [62.63; 71.04]; in the lower middle income countries group, 35.98%, 95% CI [21.89; 52.99]; in the low income countries group, 74.22%, 95% CI [54.25; 87.49] (Fig 15).

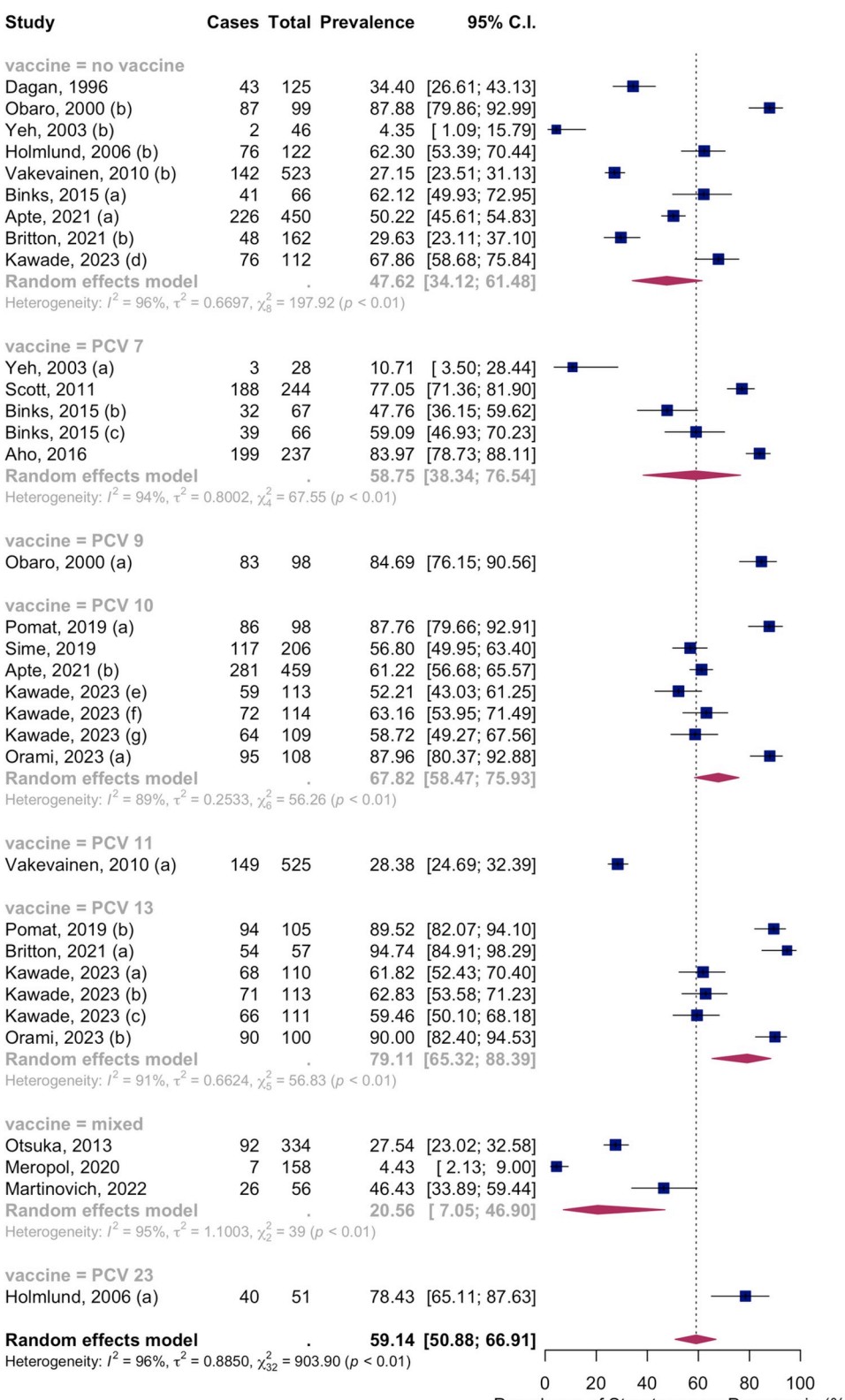

**Fig 8. Prevalence of the nasopharyngeal Streptococcus *Pneumoniae* carriage among infants at 7 to 9 months: Subgroup analysis based on vaccine type.** Abbreviations: C.I.: confidence interval.

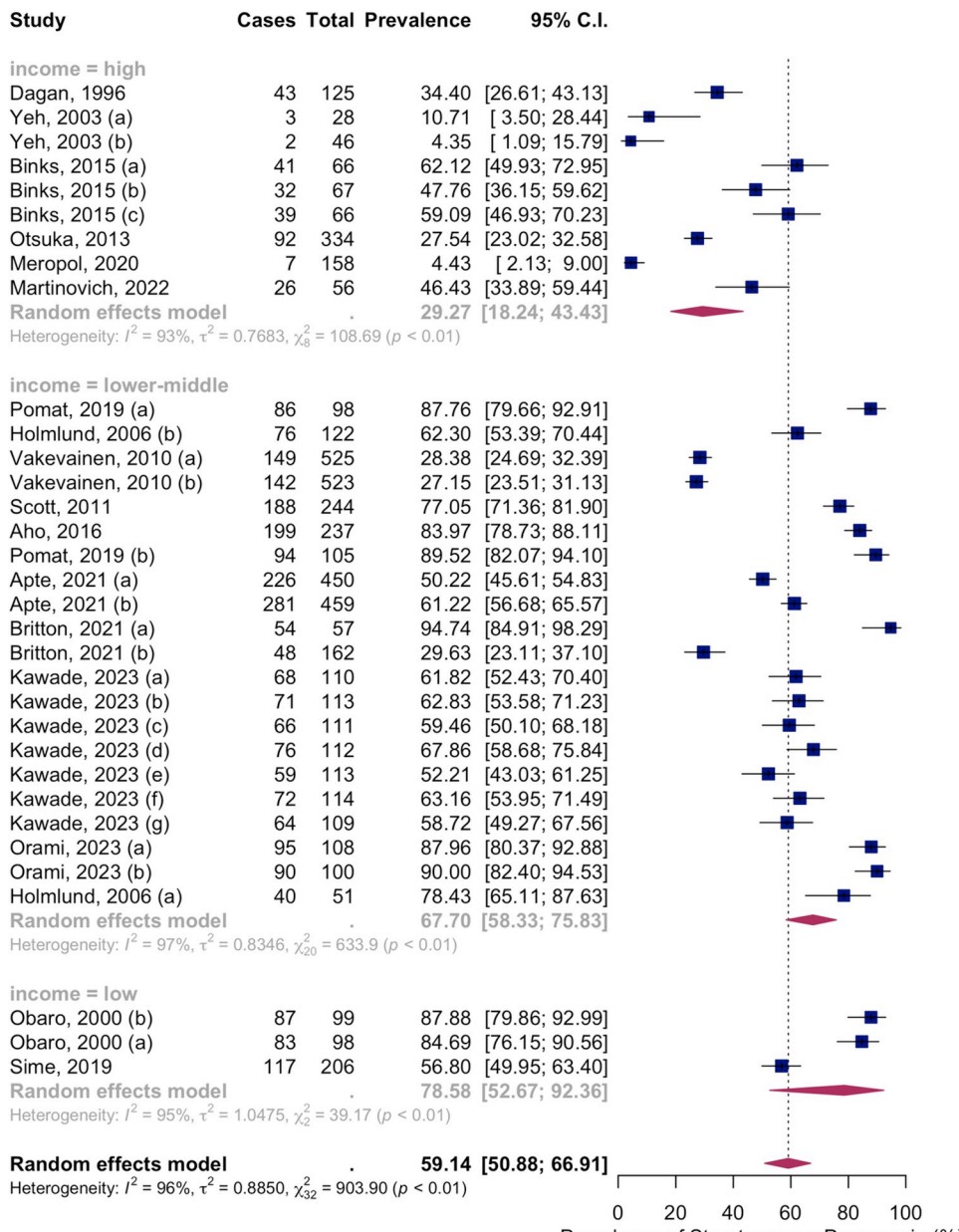

**Fig 9. Prevalence of the nasopharyngeal Streptococcus *Pneumoniae* carriage among infants at 7 to 9 months: Subgroup analysis based on country income level.** Abbreviations: C.I.: confidence interval.

## Publication bias assessment

Egger's test indicated the presence of publication bias for models in the following age groups: 0 months, 1 to 3 months, 4 to 6 months, and 13 to 18 months. Conversely, no publication bias was detected for models at 7 to 9 months, 10 to 12 months, and 19 to 24 months as presented in Table 3.

## Comparison of the NSPC rates across combined data

Comparison of the NSPC rates across combined data based on vaccination status and country income level is presented in Fig 16. The visual inspection of the bar graphs reveals that the

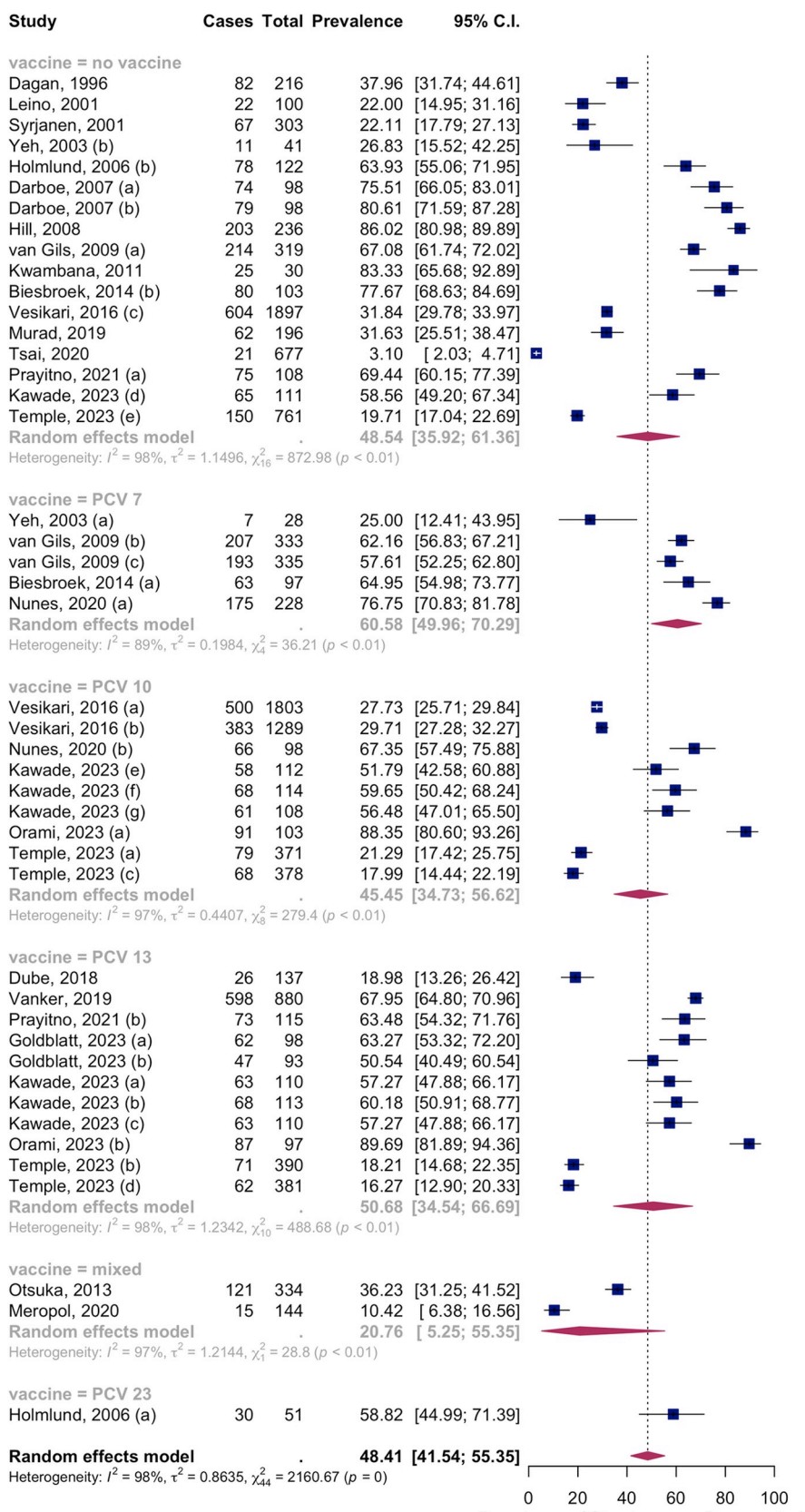

**Fig 10. Prevalence of the nasopharyngeal Streptococcus *Pneumoniae* carriage among infants at 10 to 12 months: Subgroup analysis based on vaccine type.** Abbreviations: C.I.: confidence interval.

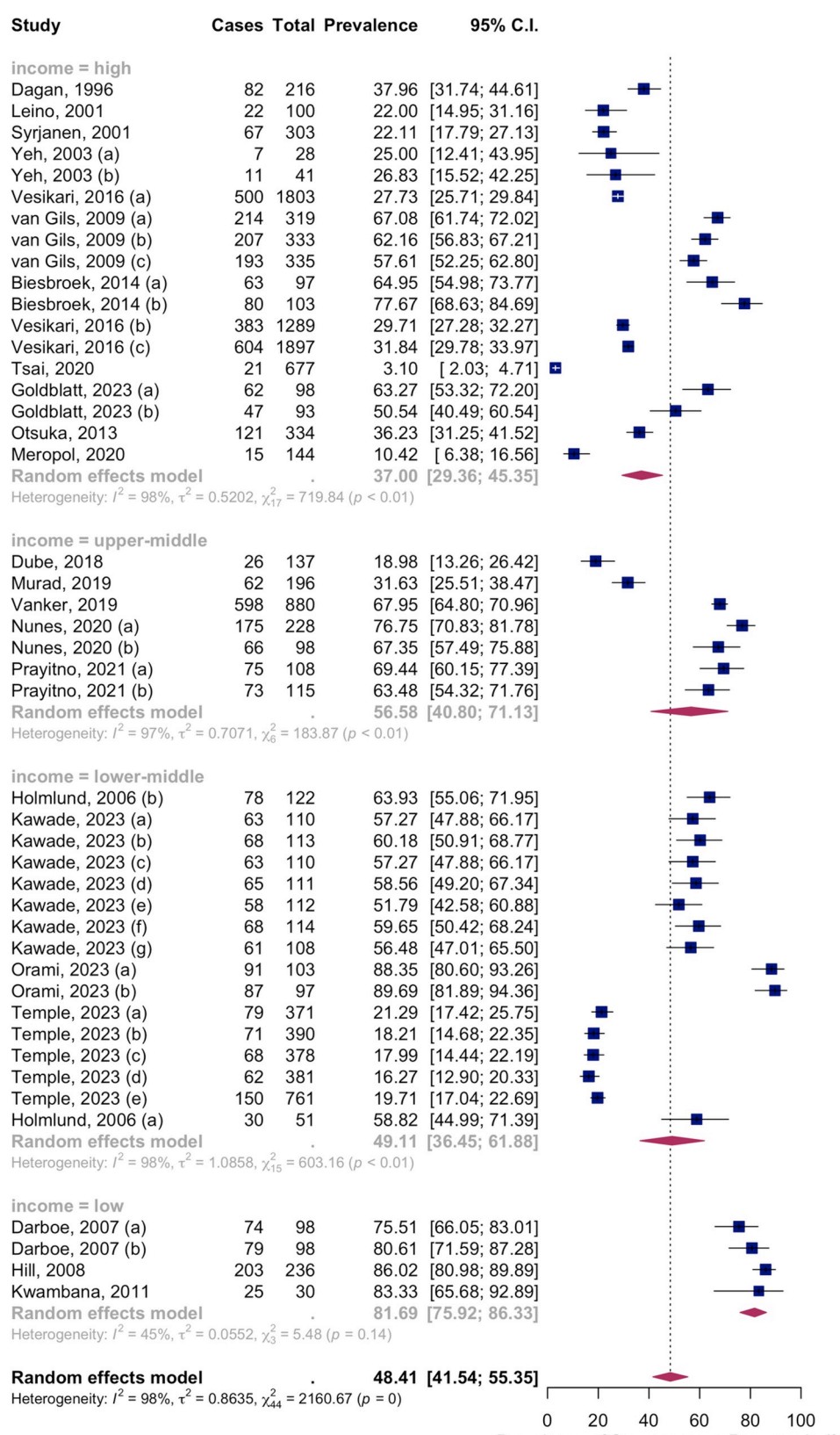

**Fig 11. Prevalence of the nasopharyngeal Streptococcus *Pneumoniae* carriage among infants at 10 to 12 months: Subgroup analysis based on country income level.** Abbreviations: C.I.: confidence interval.

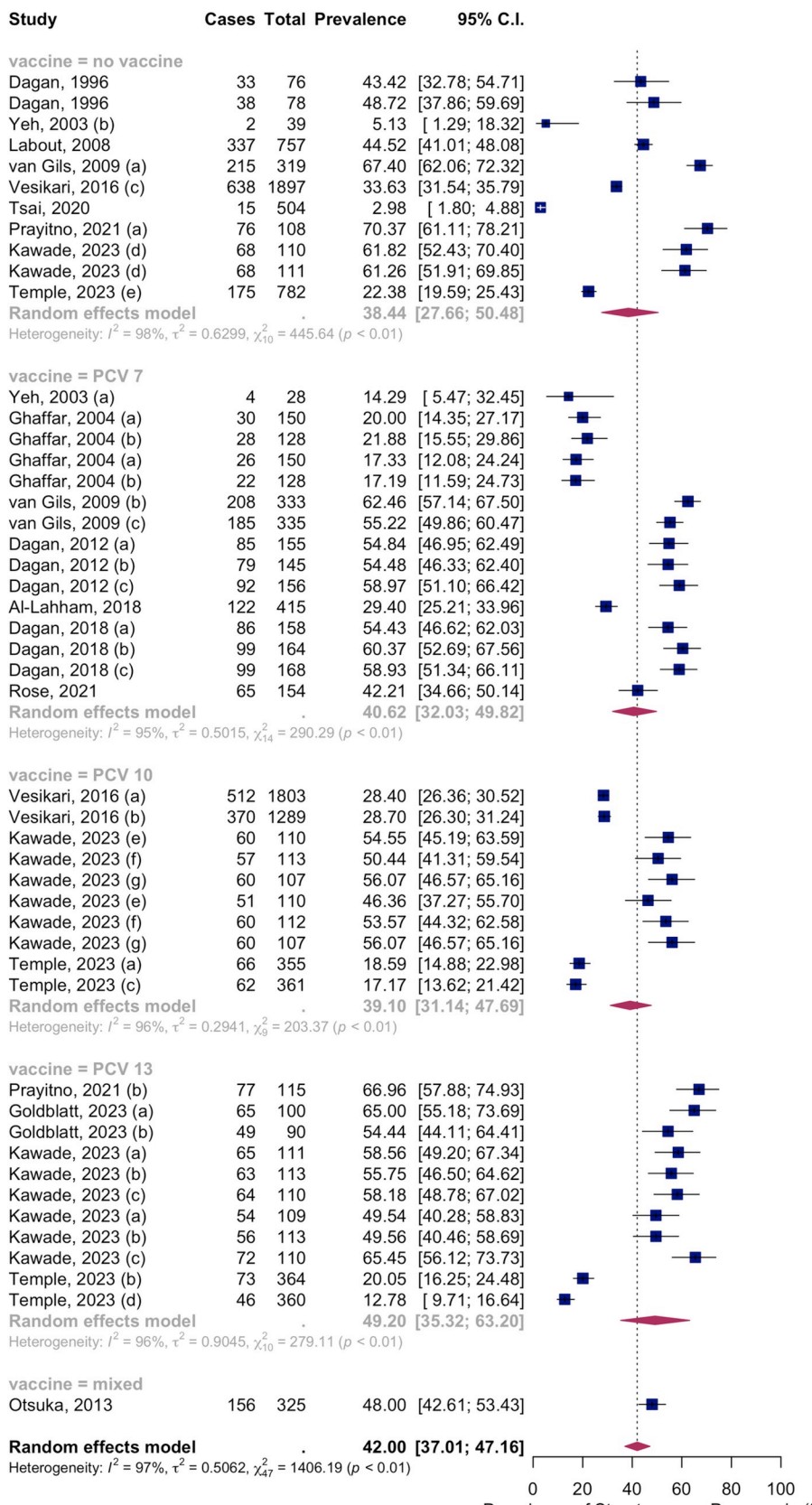

**Fig 12. Prevalence of the nasopharyngeal Streptococcus *Pneumoniae* carriage among infants at 13 to 18 months: Subgroup analysis based on vaccine type.** Abbreviations: C.I.: confidence interval.

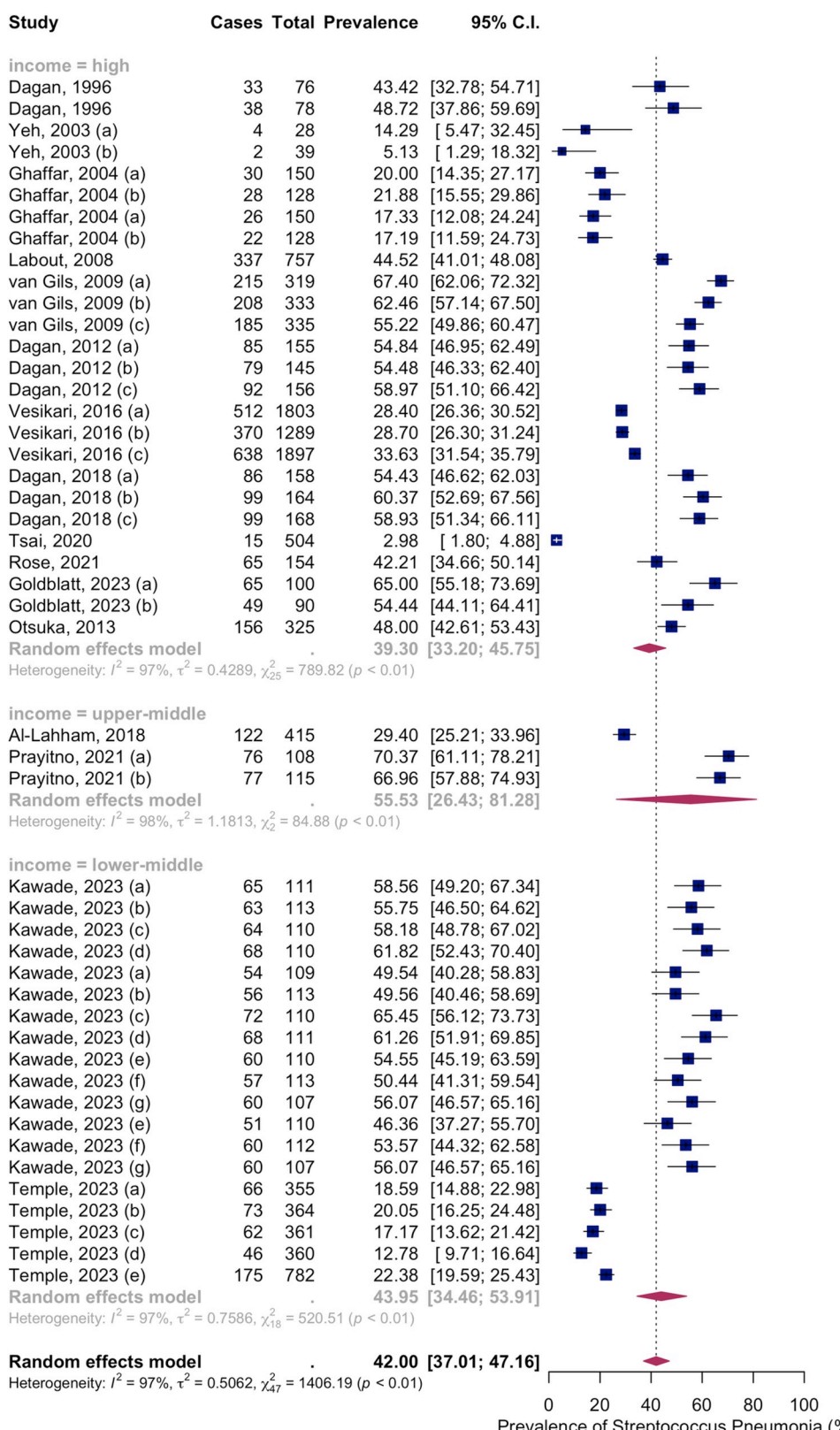

**Fig 13. Prevalence of the nasopharyngeal Streptococcus *Pneumoniae* carriage among infants at 13 to 18 months: Subgroup analysis based on country income level.** Abbreviations: C.I.: confidence interval.

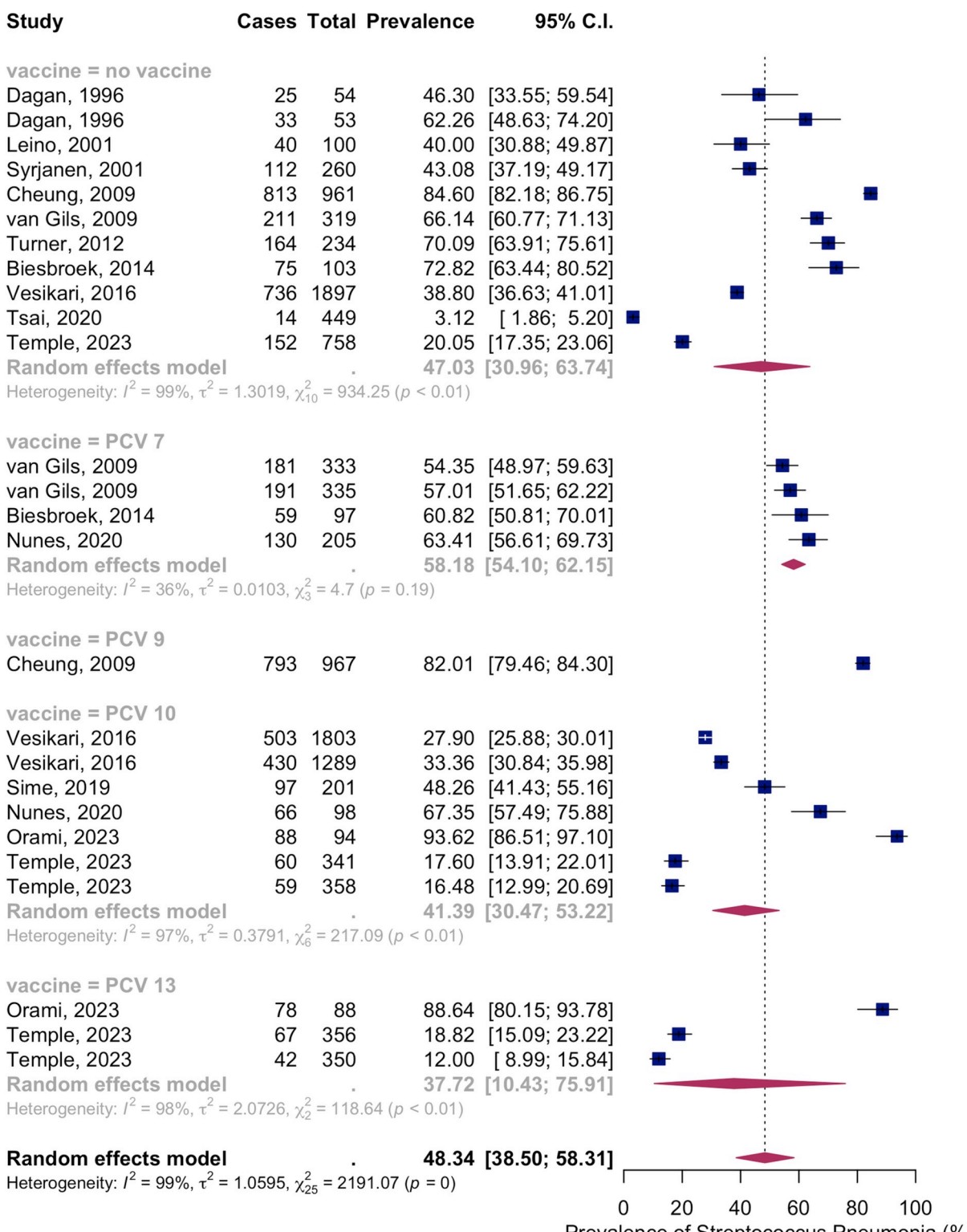

**Fig 14. Prevalence of the nasopharyngeal Streptococcus *Pneumoniae* carriage among infants at 19 to 24 months: Subgroup analysis based on vaccine type.** Abbreviations: C.I.: confidence interval.

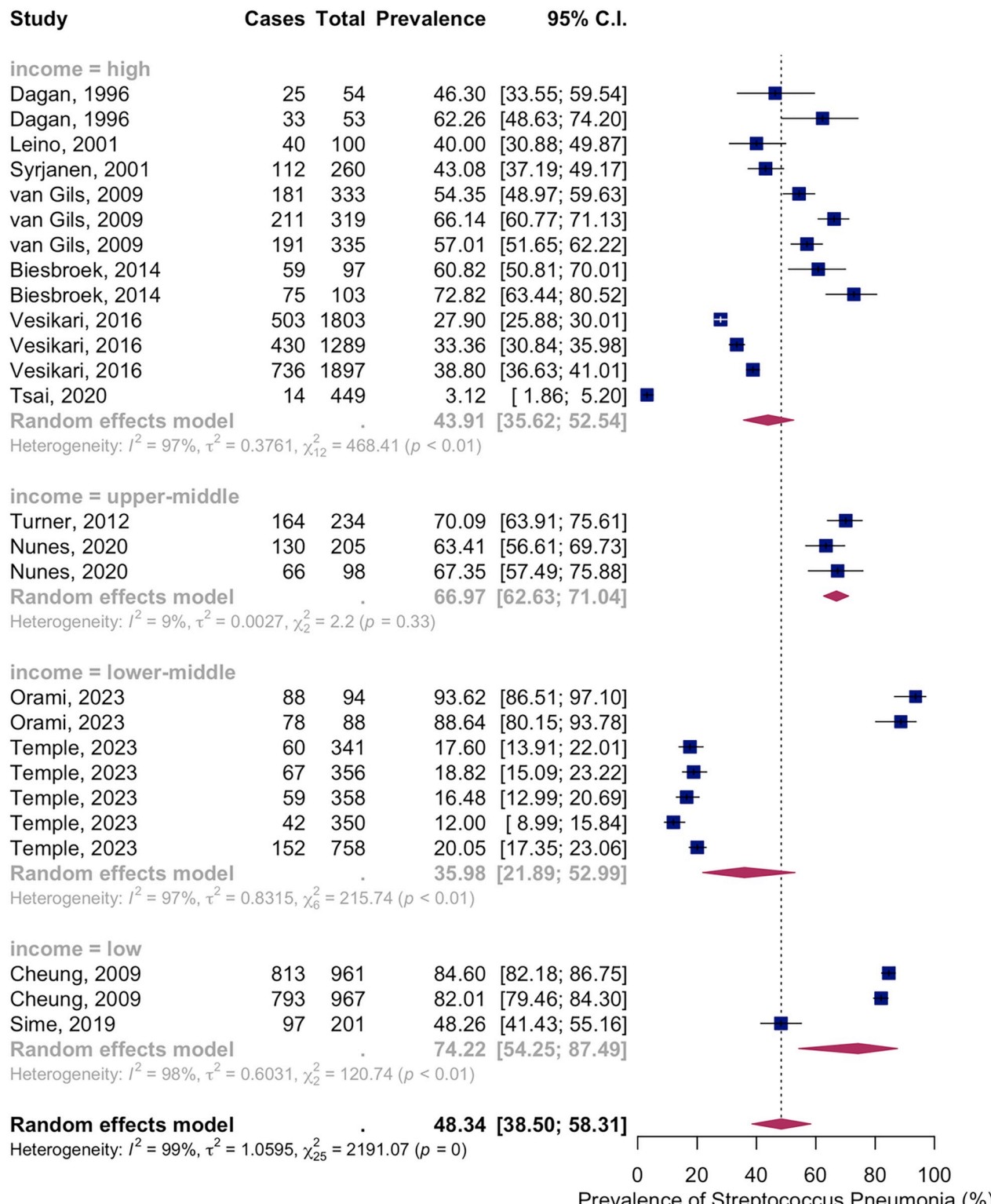

**Fig 15. Prevalence of the nasopharyngeal Streptococcus *Pneumoniae* carriage among infants at 19 to 24 months: Subgroup analysis based on country income level.** Abbreviations: C.I.: confidence interval.

**Table 3. Publication bias assessment.**

| Age | Z value | P value |
|-----|---------|---------|
| 0 months | -6.9244 | < 0.0001 |
| 1–3 months | -4.6763 | < 0.0001 |
| 4–6 months | 3.2593 | 0.0011 |
| 7–9 months | **-0.5132** | **0.6078** |
| 10–12 months | **1.7869** | **0.0740** |
| 13–18 months | -2.1412 | 0.0323 |
| 19–24 months | **1.4773** | **0.1396** |

highest NSPC rates were observed among children aged 4 to 6 months and 7 to 9 months across all groups. In the PCV7 group, the NSPC rate was higher at birth (0 months), lower at 1 to 3 months and 4 to 6 months, and then consistently higher from 7 to 9 months up to 24 months compared to the no-vaccine group. For PCV9, data are available only for the 4 to 6 months, 7 to 9 months, and 19 to 24 months age groups, all of which exhibited higher NSPC rates compared to the no-vaccine group. In the PCV10 group, the NSPC rate was higher only in the 7 to 9 months age group. For the PCV11 group, NSPC rates at 4 to 6 months and 7 to 9 months were lower than those in the corresponding age groups of the no-vaccine group. In the PCV13 group, the NSPC rate was lower at birth (0 months) and at 19 to 24 months, while it was higher in all other age groups compared to the no-vaccine group. For the PCV23 group, data are available for the 1 to 3 months, 4 to 6 months, 7 to 9 months, and 10 to 12 months age groups, with higher NSPC rates observed in all these groups except the 4 to 6 months group. In the mixed vaccine group, there were no data for the 19 to 24 months age group, and the only age group with a higher NSPC rate compared to the no-vaccine group was the 13 to 18 months age group.

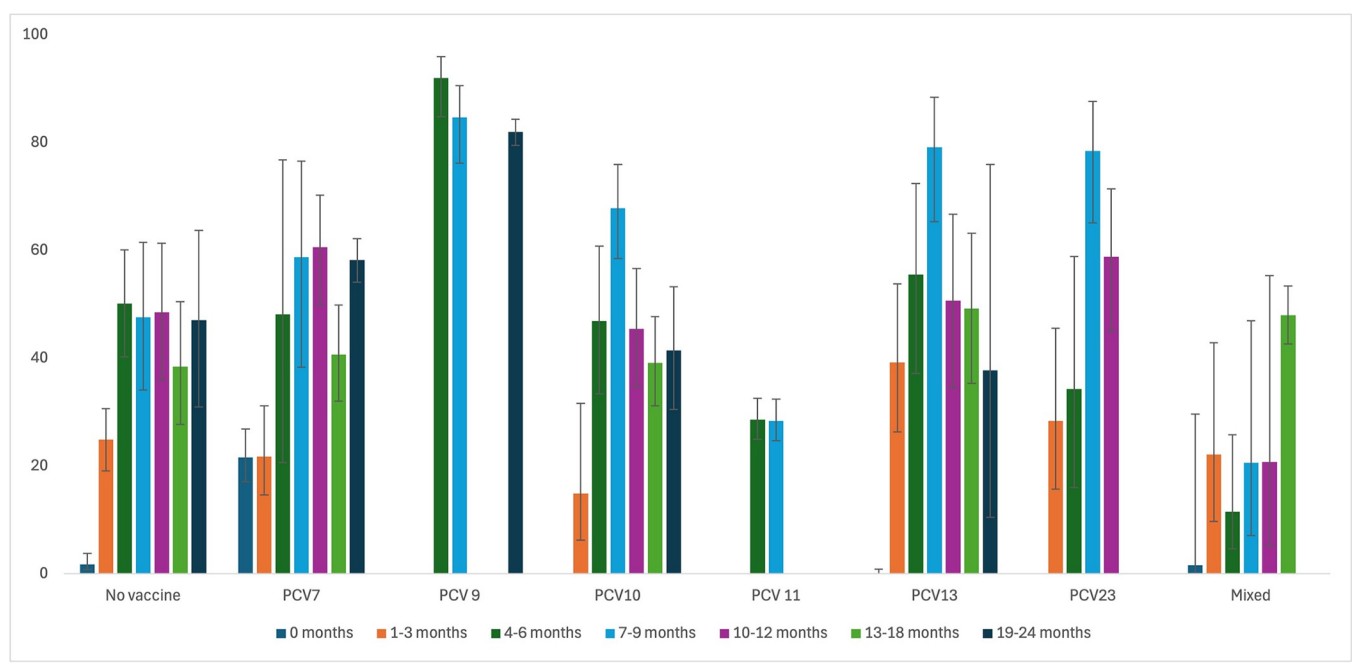

**Fig 16. Nasopharyngeal Streptococcus *Pneumoniae* carriage changes: Subgroup analysis based on vaccine type.**

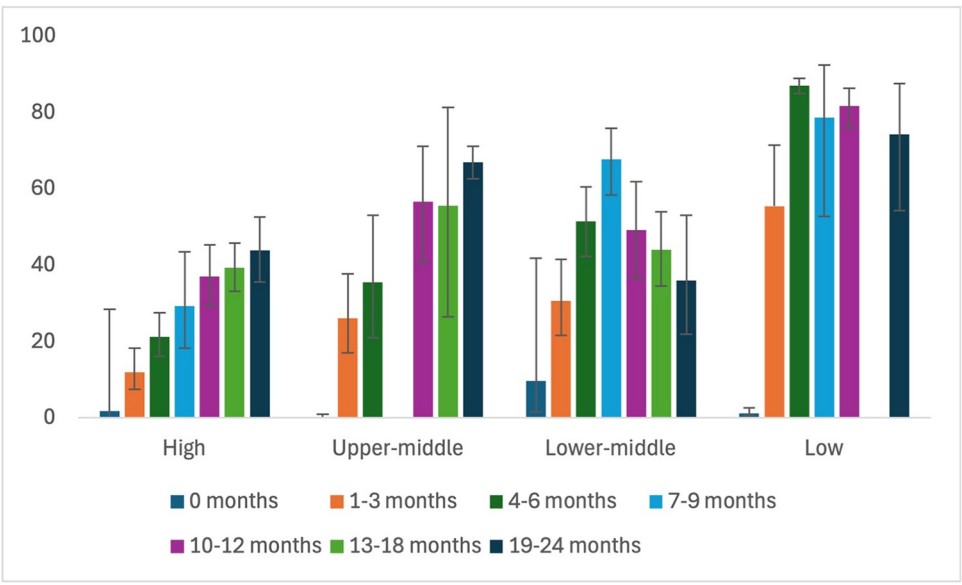

**Fig 17. Nasopharyngeal Streptococcus *Pneumoniae* carriage changes: Subgroup analysis based on country income level.**

The bar graph, synthesizing data from a meta-analysis stratified by countries' income levels, elucidates notable patterns in subgroup analyses (Fig 17). Specifically, across all age groups examined, countries categorized as high income consistently exhibit the lowest rates of NSPC compared to their counterparts in upper-middle, lower-middle-, and low-income categories. In contrast, low-income countries consistently demonstrate the highest NSPC rates across all age categories studied.

## Discussion

This systematic review and meta-analysis provide a comprehensive assessment of the prevalence of NSPC in healthy infants during their first two years of life. Our findings reveal a notable prevalence of NSPC across diverse vaccine groups and income settings, indicating a sustained rate of pneumococcal carriage in this vulnerable age group. The pooled prevalence from cohort studies and randomized controlled trials suggests that NSPC remains prevalent in infants aged 4 to 6 months and 7 to 9 months, irrespective of vaccination status, with a general decline observed afterward.

Vaccination has profoundly impacted pneumococcal serotype carriage, with the introduction of PCVs leading to a significant reduction in vaccine serotypes among infants [70]. However, non-vaccine serotypes continue to persist, albeit generally with lower virulence, and still contribute to NSPC, underscoring the need for ongoing serotype surveillance due to serotype replacement. The overall reduction in vaccine-type carriage represents a public health milestone in lowering invasive IPD risks, such as pneumonia, meningitis, and sepsis, in vaccinated populations [71].

The evolution of PCV formulations has aimed to address serotype replacement. Following the WHO's recommendation in 2010 to transition from PCV7 to PCV13, which includes additional serotypes (1, 3, 5, 6A, 7F, and 19A), significant strides were made in reducing IPD. However, as non-vaccine serotypes emerge and become more virulent, the development and potential widespread implementation of higher-valent vaccines, such as the 20-valent PCV

(PCV20), is actively under consideration. Studies from high- and upper-middle-income countries indicate that PCV20, covering additional serotypes (8, 10A, 11A, 12F, 15B, 22F, and 33F), is not only effective but also cost-efficient, especially in pediatric populations using the 2+1 dosing schedule [72–78]. Recent systematic reviews suggest that transitioning from PCV13 to PCV20, rather than to the 15-valent PCV (PCV15), would result in greater cost savings by effectively addressing serotype replacement and reducing the economic burden associated with pneumococcal diseases [79].

Our analysis highlights significant disparities in NSPC based on country income level. Infants in low-income countries exhibit higher NSPC rates, with 74.22% (95% CI [54.25; 87.49]) prevalence among 19- to 24-month-olds. Factors such as limited vaccination coverage, crowded living conditions, and socio-economic constraints contribute to this elevated carriage rate. Studies, including work by Adegbola and colleagues, show a pre-vaccine NSPC prevalence as high as 93.4% in children up to 48 months in low-income settings [80]. The pooled prevalence estimates from in this analysis further illustrate these disparities: 64.8% in low-income and 47.8% in lower-middle-income countries [80], underscoring the need for tailored healthcare policies to address NSPC in resource-limited settings.

The findings of our analysis indicate that younger age, poverty, and cohabitation with other young children are associated with increased NSPC, especially in low-income countries. Addressing these disparities requires focused healthcare policies and vaccination strategies to mitigate NSPC rates and improve pneumococcal disease prevention. Ongoing surveillance and cost-benefit evaluations of broader PCV20 implementation could inform future vaccine strategies and ensure continued protection against both vaccine and non-vaccine serotypes.

Several limitations of this meta-analysis must be noted. First, the variability in study designs, sample sizes, and geographical contexts introduces heterogeneity that may impact the findings' generalizability. Additionally, differences in laboratory methods for NSPC detection across studies could influence reported prevalence rates. Lastly, this review did not analyze serotype distribution within Streptococcus *pneumoniae*, which is crucial for future vaccine strategy development. Understanding the prevalence of specific serotypes is essential to tailoring pneumococcal vaccine formulations effectively, especially for low-income regions where serotype diversity may significantly influence vaccine effectiveness.

## Conclusion

This systematic review underscores the high prevalence of NSPC in infants, with marked country income-level disparities. Continuous monitoring of NSPC and emerging non-vaccine serotypes is essential for refining vaccination strategies and reducing the overall burden of pneumococcal diseases in infants. Future research should prioritize longitudinal studies to enhance our understanding of NSPC dynamics over time and the long-term efficacy of current and emerging vaccines.

## Supporting information

**S1 Checklist. PRISMA 2020 checklist.**
(DOCX)

**S1 Table. Search strategy.**
(DOCX)

**S2 Table. Detailed study selection process results.**
(XLSX)

**S3 Table. Comprehensive data extraction summary.**
(XLSX)

## Author Contributions

**Conceptualization:** Gulzhan Beissegulova.

**Data curation:** Bakyt Ramazanova, Aliya Mamatova.

**Formal analysis:** Gulzhan Beissegulova, Bibigul Seitkhanova, Zhaksylyk Seiitbay.

**Investigation:** Gulzhan Beissegulova, Bibigul Seitkhanova, Zhaksylyk Seiitbay.

**Methodology:** Gulzhan Beissegulova, Zhaksylyk Seiitbay.

**Project administration:** Yekaterina Koloskova, Ratbek Sailaubekuly.

**Resources:** Bakyt Ramazanova, Bibigul Seitkhanova.

**Software:** Bakyt Ramazanova, Aliya Mamatova.

**Supervision:** Bakyt Ramazanova, Tolkyn Begadilova, Ulzhan Iskakova.

**Validation:** Kamilya Mustafina, Ratbek Sailaubekuly.

**Visualization:** Bibigul Seitkhanova.

**Writing – original draft:** Gulzhan Beissegulova, Tolkyn Begadilova, Yekaterina Koloskova, Aliya Mamatova, Zhaksylyk Seiitbay.

**Writing – review & editing:** Gulzhan Beissegulova, Bakyt Ramazanova, Kamilya Mustafina, Bibigul Seitkhanova, Ulzhan Iskakova, Ratbek Sailaubekuly.

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
