## [Decision Letter · Decision Letter 0]

23 Oct 2024

PONE-D-24-29372Prevalence of Nasopharyngeal Streptococcus Pneumonia Carriage in Infants: A Systematic Review and Meta-Analysis of Cohort Studies and Randomized Controlled TrialsPLOS ONE

Dear Dr. Beissegulova,

Thank you for submitting your manuscript to PLOS ONE. After careful consideration, we feel that it has merit but does not fully meet PLOS ONE’s publication criteria as it currently stands. Therefore, we invite you to submit a revised version of the manuscript that addresses the points raised during the review process.

Please pay attention to the comments made my reviewers. Please ensure that data in table and text match. I agree with the Reviewers that the discussion should be more focused and coherent.

We look forward to receiving your revised manuscript.

Kind regards,

Anirudh K. Singh, Ph.D

Academic Editor

PLOS ONE

2. In the online submission form, you indicated that [Data used for the analysis in this study is available from the corresponding author upon reasonable request].

3. As required by our policy on Data Availability, please ensure your manuscript or supplementary information includes the following:

Reviewers' comments:

Reviewer's Responses to Questions

**Comments to the Author**

1. Is the manuscript technically sound, and do the data support the conclusions?

Reviewer #1: Yes

Reviewer #2: Yes

2. Has the statistical analysis been performed appropriately and rigorously? 

Reviewer #1: I Don't Know

Reviewer #2: Yes

3. Have the authors made all data underlying the findings in their manuscript fully available?

Reviewer #1: Yes

Reviewer #2: Yes

4. Is the manuscript presented in an intelligible fashion and written in standard English?

Reviewer #1: Yes

Reviewer #2: Yes

5. Review Comments to the Author

Reviewer #1: This systematic review approaches the pneumococcal carriage state for a much younger age group with very interesting results. The manuscript will benefit from significant shortening especially the results section for figures. My main criticism would be the language highly equating the carriage with clinical illness. Especially in infants less than 6 months of age the clinical pneumococcal disease is not significant.

Under methods, please specify the analysis period. Please also provide references for the statistical methods used. Please also explain how the age categories were calculated based on the review as some publications may not have reported this.

Reviewer #2: In this systematic review and meta-analysis, the authors determined the prevalence of nasopharyngeal carriage of Streptococcus pneumoniae (NSPC) among infants during their first two years of life, stratified by month of age. They also did subgroup analysis for the carriage rates among infants who received different vaccines and also based on country income levels. The authors reported that the pooled mean prevalence of NSPC was highest in the 7 to 9-month age group. Additionally, the prevalence was higher among infants aged 4 to 6 months and 7 to 9 months, across all vaccine groups. Furthermore, the study found that low-income countries exhibited the highest NSPC rates across all age groups.

• Although the manuscript has considered all the necessary parameters required for performing meta-analysis like PRISMA guidelines, risk of bias assessment, publication bias assessment, however, it needs to be better structured so that one can follow it easily.

• In the title, it is Streptococcus pneumoniae carriage, “e” is missing from pneumoniae. Moreover, it should be in italics.

• In material and methods section, authors should mention PICOS in the search strategy. Authors have mentioned risk of bias, instead they can add study quality assessment. Moreover, authors have not mentioned anything about the restrictions imposed, if any.

• In result section, in line no. 207, India and Bangladesh should be removed as the no. are not getting matched, India has got repeated and Bangladesh is not in the table.

• In the result section, Data on the NSPC at birth comprises of vaccine groups which includes PCV 7 and PCV 10 group in line no. 238, while in the diagram it is PCV 7 and PCV 13. Please clarify.

• In the subgroup analysis of the vaccine group, authors should talk about dosage of the vaccination as well as serotypes that are covered. Infact, in my opinion authors have categorized groups taking dosage into account. If that is the case, it should be more elaborative and clear. If not than they should specify how they have categorized the groups. Authors should add that in supplementary information.

• Discussion section is poorly written, it needs to be more structured and organized. Authors should discuss about the vaccine type which is more effective and serotypes covered by the vaccines. Also whether the observed prevalence of NSPC is due to vaccine or non-vaccine type serotypes.

6. PLOS authors have the option to publish the peer review history of their article (what does this mean?). If published, this will include your full peer review and any attached files.

Reviewer #1: No

Reviewer #2: No

---

## [Author Response · Author response to Decision Letter 0]

2 Nov 2024

Response to the reviewers

Authors reply: The manuscript and the title page have been revised to follow the journal guidelines. The author information was deleted from the first page, and the title of the manuscript was revised.

2. In the online submission form, you indicated that [Data used for the analysis in this study is available from the corresponding author upon reasonable request].

Authors reply: All requested data is provided as Supplementary Tables (S2 Table and S3 Table).

3. As required by our policy on Data Availability, please ensure your manuscript or supplementary information includes the following:

Authors reply: all requested data that were missing from the manuscript are provided as Supplementary Tables (S2 Table and S3 Table).

We have also revised the data availability statement, and added the S2 Table and S3 Table Titles in lines 520 – 521: S2 Table. Detailed study selection process results

S3 Table. Comprehensive data extraction summary

Lines 542 – 544: Availability of data

All data necessary to replicate the study’s findings is provided in supplemental materials.

Reviewer #1: 

1. This systematic review approaches the pneumococcal carriage state for a much younger age group with very interesting results. The manuscript will benefit from significant shortening especially the results section for figures. My main criticism would be the language highly equating the carriage with clinical illness. Especially in infants less than 6 months of age the clinical pneumococcal disease is not significant.

Authors reply: 

The authors sincerely appreciate the reviewer’s insightful feedback regarding our manuscript. We recognize the importance of clarity in differentiating between pneumococcal carriage and clinical illness, particularly for infants under six months, where the incidence of clinical disease is indeed low.

We have taken steps to address your concerns by substantially shortening the results section and restructuring the presentation to improve readability and coherence. The number of figures presented is integral to our objective of exploring the prevalence of non-invasive pneumococcal carriage (NSPC) among infants in their first two years, while also considering variations by vaccine type and country income levels.

In our revisions, we have highlighted significant disparities in NSPC prevalence across different economic contexts, emphasizing the need for a nuanced understanding of these factors, and have discussed this issue in more detail in the revised discussion section. All modifications have been clearly marked in the results and discussion sections of the manuscript, particularly between lines 219 - 389 & 442 -508. We hope that these changes will satisfactorily address the reviewer’s comments and enhance the overall clarity and impact of our work.

2. Under methods, please specify the analysis period. Please also provide references for the statistical methods used. Please also explain how the age categories were calculated based on the review as some publications may not have reported this.

Authors reply: The authors appreciate the reviewer’s valuable feedback, and offer the following changes:

1. We have added the following statement in lines 103 – 104 to specify the analysis period: No restrictions were imposed on the publication dates of studies

2. The following reference was added to the statistical methods: Subgroup analysis based on vaccination status and country income level was used to calculate the pooled mean prevalence of NSPC for each specified age group, along with 95% confidence intervals (95% CI), using a random-effects model for meta-analysis in RStudio software with the “meta” and “metafor” packages [18]. 

18. Harrer M, Cuijpers P, Furukawa TA, Ebert DD. Doing Meta-Analysis with R: A Hands-On Guide. Boca Raton, FL and London: Chapman & Hall/CRC Press.; 2021. Available: https://bookdown.org/MathiasHarrer/Doing_Meta_Analysis_in_R/pooling-es.html

3.The authors appreciate the reviewer’s valuable feedback regarding the categorization of age groups. We have derived the age categories presented in this review based on a consensus among the three authors responsible for data extraction, after a careful review of existing literature. In instances where studies reported NSPC rates in age brackets that differed significantly from those used in our meta-analysis (e.g., 3–12 months or 12–36 months), we excluded these studies to maintain consistency across age groupings. A summary of all excluded articles with non-aligned age ranges is provided in Supplemental Table S2 for further clarity. We hope this approach clarifies the methodology used for age categorization and meets the reviewer’s expectations.

Reviewer #2: In this systematic review and meta-analysis, the authors determined the prevalence of nasopharyngeal carriage of Streptococcus pneumoniae (NSPC) among infants during their first two years of life, stratified by month of age. They also did subgroup analysis for the carriage rates among infants who received different vaccines and also based on country income levels. The authors reported that the pooled mean prevalence of NSPC was highest in the 7 to 9-month age group. Additionally, the prevalence was higher among infants aged 4 to 6 months and 7 to 9 months, across all vaccine groups. Furthermore, the study found that low-income countries exhibited the highest NSPC rates across all age groups.

1. Although the manuscript has considered all the necessary parameters required for performing meta-analysis like PRISMA guidelines, risk of bias assessment, publication bias assessment, however, it needs to be better structured so that one can follow it easily.

Authors reply: the authors acknowledge the reviewer’s insightful feedback. The results and discussion section have been extensively revised, and in the results section, we have added the explanatory paragraph on the structure of the results presentation in lines 219 – 223: In this section, we present the prevalence of nasopharyngeal SP carriage in infants during the first two years of life, stratified by vaccine type and country income level. The data are organized by age intervals as follows: 0 months, 1–3 months, 4–6 months, 7–9 months, 10–12 months, 13–18 months, and 19–24 months.

2. In the title, it is Streptococcus pneumoniae carriage, “e” is missing from pneumoniae. Moreover, it should be in italics.

Authors reply: the authors acknowledge the reviewer’s comment and agree with the reviewer. The revised manuscript title is as follows: Prevalence of nasopharyngeal Streptococcus pneumoniae carriage in infants: a systematic review and meta-analysis of cohort studies and randomized controlled trials

3. In material and methods section, authors should mention PICOS in the search strategy. Authors have mentioned risk of bias, instead they can add study quality assessment. Moreover, authors have not mentioned anything about the restrictions imposed, if any.

Authors reply: the authors acknowledge the reviewer’s thoughtful comments and agree with the reviewer. 

1. The authors have revised the search strategy description, and have added the following description in lines 94 – 100: The search was structured according to the Population, Intervention, Comparator, Outcomes, and Study Design (PICOS) framework as follows: Population (P): infants and children under two years of age; Intervention (I): not applicable; Comparator (C): not applicable; Outcomes (O): nasopharyngeal carriage; and Study Design (S): cohort studies and randomized clinical trials (RCTs).

2. The risk of bias assesment and the assessment of the quality of the included studies were performed by the same tool. The authors changed the subheading to reflect this important omission in line 146: Risk of bias and study quality assessment

3. Restrictions imposed to the search strategy are provided in the Full search strategy as Supplemental Table.

4. In result section, in line no. 207, India and Bangladesh should be removed as the no. are not getting matched, India has got repeated and Bangladesh is not in the table.

Authors reply: The authors acknowledge the reviewer’s thoughtful comment and agree with the reviewer. The study by Apte, 2021 provided data for children in India and Bangladesh, and the Table 1 was missing this information. We have revised the Table 1 and added the Bangladesh to the list.

5. In the result section, Data on the NSPC at birth comprises of vaccine groups which includes PCV 7 and PCV 10 group in line no. 238, while in the diagram it is PCV 7 and PCV 13. Please clarify.

Authors reply: The authors acknowledge the reviewers thoughtful comment, and agree with the reviewer. This was an omitted typo from the authors side, the groups description was changed to PCV7 and PCV13, as reflected in the Figure 2. The complete data extraction table is added to the manuscript as supplemental S3 Table, and the data is also provided there to support the clarification.

6. In the subgroup analysis of the vaccine group, authors should talk about dosage of the vaccination as well as serotypes that are covered. Infact, in my opinion authors have categorized groups taking dosage into account. If that is the case, it should be more elaborative and clear. If not than they should specify how they have categorized the groups. Authors should add that in supplementary information.

Authors reply: The authors acknowledge the reviewer’s thoughtful comments and agree with the reviewer. The dosage of the vaccination was also extracted as presented in S3 Table (data extraction table), but the data was too heterogeneous and incomplete to analyze the vaccination based on the dosage. Thus, we did not include any information on the analysis based on the vaccination dosage to the methods section.

We have also expanded the limitation section of the manuscript to reflect that we did not investigate the serotypes in Lines 503 – 508: Lastly, this review did not analyze serotype distribution within Streptococcus pneumoniae, which is crucial for future vaccine strategy development. Understanding the prevalence of specific serotypes is essential to tailoring pneumococcal vaccine formulations effectively, especially for low-income regions where serotype diversity may significantly influence vaccine effectiveness.

7. Discussion section is poorly written, it needs to be more structured and organized. Authors should discuss about the vaccine type which is more effective, and serotypes covered by the vaccines. Also, whether the observed prevalence of NSPC is due to vaccine or non-vaccine type serotypes.

Authors reply: The authors acknowledge the reviewer’s thoughtful comments and agree with the reviewer. We have revised the discussion to follow the following outline: main findings of the study, vaccine effectiveness and serotype dynamics, disparities in NSPC prevalence, policy implications, limitations and conclusion as presented in the updated discussion section. The changes are highlighted starting from line 440 – 508:

This systematic review and meta-analysis provide a comprehensive assessment of the prevalence of NSPC in healthy infants during their first two years of life. Our findings reveal a notable prevalence of NSPC across diverse vaccine groups and income settings, indicating a sustained rate of pneumococcal carriage in this vulnerable age group. The pooled prevalence from cohort studies and randomized controlled trials suggests that NSPC remains prevalent in infants aged 4 to 6 months and 7 to 9 months, irrespective of vaccination status, with a general decline observed afterward.

Vaccination has profoundly impacted pneumococcal serotype carriage, with the introduction of PCVs leading to a significant reduction in vaccine serotypes among infants [69]. However, non-vaccine serotypes continue to persist, albeit generally with lower virulence, and still contribute to NSPC, underscoring the need for ongoing serotype surveillance due to serotype replacement. The overall reduction in vaccine-type carriage represents a public health milestone in lowering invasive IPD risks, such as pneumonia, meningitis, and sepsis, in vaccinated populations [70].

The evolution of PCV formulations has aimed to address serotype replacement. Following the WHO’s recommendation in 2010 to transition from PCV7 to PCV13, which includes additional serotypes (1, 3, 5, 6A, 7F, and 19A), significant strides were made in reducing IPD. However, as non-vaccine serotypes emerge and become more virulent, the development and potential widespread implementation of higher-valent vaccines, such as the 20-valent PCV (PCV20), is actively under consideration. Studies from high- and upper-middle-income countries indicate that PCV20, covering additional serotypes (8, 10A, 11A, 12F, 15B, 22F, and 33F), is not only effective but also cost-efficient, especially in pediatric populations using the 2+1 dosing schedule [71–77]. Recent systematic reviews suggest that transitioning from PCV13 to PCV20, rather than to the 15-valent PCV (PCV15), would result in greater cost savings by effectively addressing serotype replacement and reducing the economic burden associated with pneumococcal diseases [78].

Our analysis highlights significant disparities in NSPC based on country income level. Infants in low-income countries exhibit higher NSPC rates, with 74.22% (95% CI [54.25; 87.49]) prevalence 

---

## [Editor Report · Decision Letter 1]

26 Nov 2024

Prevalence of nasopharyngeal Streptococcus Pneumoniae carriage in infants: a systematic review and meta-analysis of cohort studies and randomized controlled trialsls

PONE-D-24-29372R1

Dear Dr. Beissegulova,

We’re pleased to inform you that your manuscript has been judged scientifically suitable for publication and will be formally accepted for publication once it meets all outstanding technical requirements.

Kind regards,

Anirudh K. Singh, Ph.D

Academic Editor

PLOS ONE
---

## [Editor Report · Acceptance letter]

3 Dec 2024

PONE-D-24-29372R1 

PLOS ONE

Dear Dr. Beissegulova, 

I'm pleased to inform you that your manuscript has been deemed suitable for publication in PLOS ONE. Congratulations! Your manuscript is now being handed over to our production team.

Kind regards, 

on behalf of

Dr. Anirudh K. Singh 

Academic Editor

PLOS ONE